



# Proglacial Lakes Elevate Glacier Surface Velocities in the Himalayan Region

Jan B. Pronk[1], Tobias Bolch[1], Owen King[1], Bert Wouters[2], Douglas I. Benn[1]

[1]School of Geography and Sustainable Development, University of St Andrews, St Andrews, UK
[2]Institute for Marine and Atmospheric Research Utrecht, Utrecht University, the Netherlands

*Correspondence to:* Jan B. Pronk (janboukepronk@gmail.com)

**Abstract.** Meltwater from Himalayan glaciers sustains the flow of rivers such as the Ganges and Brahmaputra on which over half a billion people depend for day-to-day needs. Upstream areas are likely to be affected substantially by climate change, and changes in the magnitude and timing of meltwater supply are likely to occur in coming decades. About 10 % of the
Himalayan glacier population terminates into pro-glacial lakes and such lake-terminating glaciers are known to exhibit higher than average total mass losses. However, relatively little is known about the mechanisms driving exacerbated ice loss from lake-terminating glaciers in the Himalaya. Here we examine a composite (2017-2019) glacier surface velocity dataset, derived from Sentinel 2 imagery, covering Central and Eastern Himalayan glaciers larger than 3 km$^2$. We find that centre flow line velocities of lake-terminating glaciers are more than double those of land-terminating glaciers (18.8 vs 8.24 m yr$^{-1}$) and show
substantially more heterogeneity around glacier termini. We attribute this large heterogeneity to the varying influence of lakes on glacier dynamics, resulting in differential rates of dynamic thinning, which effects about half of the clean-ice lake-terminating glacier population. Numerical ice-flow model experiments show that changes at the frontal boundary condition are likely to play a key role in accelerating the glacier flow at the front, with variations in basal friction only being of modest importance. The expansion of current glacial lakes, and the formation of new meltwater bodies will influence the dynamics of
an increasing number of Himalayan glaciers in the future; a scenario not currently considered in regional ice loss projections.



## 1 Introduction

Himalayan glaciers provide an important baseline supply of meltwater for downstream areas (Bolch, 2017; Immerzeel et al., 2010; Pritchard, 2019; Viviroli et al., 2007). A large decrease in runoff from the rivers that drain this mountain range will have

major implications for downstream water security, particularly in the populous catchments of the Ganges, Indus, and Brahmaputra rivers. Although a drastic reduction in glacier area and mass is projected in the Himalaya over the 21st century (Kraaijenbrink et al., 2017; Rounce et al., 2020), large uncertainties in the pace of the loss exist (Lutz et al., 2013). Hence, there are also large uncertainties in future melt water supply, and an improved understanding of the evolution of Himalayan glaciers is needed.

Himalayan glaciers have been retreating and losing mass since the mid-19th century and rates of mass loss have been increasing over at least the last four decades (Bolch et al., 2012; Brun et al., 2017; King et al., 2019; Maurer et al., 2019). Various studies report Himalayan averaged glacier mass losses of around $-0.40 \pm 0.10$ m.w.e. $yr^{-1}$ since the beginning of this century (Brun et al., 2017; Shean et al., 2020), which roughly translates into a total mass loss of 7.5 Gt $yr^{-1}$. However, within the Himalayan mountains, large intra-regional variability in glacier mass loss exists (Azam et al., 2018; Bolch et al., 2012; Brun et al., 2017;

King et al., 2019; Maurer et al., 2019), which indicates that there are factors capable of exacerbating -or reducing- glacial mass losses that are at least partially decoupled from climate.

The development of proglacial lakes in direct contact with the glacier terminus has been linked with enhanced glacier mass loss in the Himalayan region (King et al., 2019; Maurer et al., 2019). This contrast in mass loss with land-terminating glaciers manifests itself in two ways, namely by elevated terminal retreat rates and by enhanced surface lowering towards the terminus

of the lake-terminating glaciers (King et al., 2019). The latter indicates that proglacial lakes can influence the flow characteristics of their host glacier through dynamic thinning.

A factor that further complicates the dynamics and mass loss rate of lake-terminating glaciers in the Himalayas is the presence of a thick layer of debris, which is widespread on Himalayan glaciers (Herreid and Pellicciotti, 2020). The low-gradient, debris-covered portions of many Himalayan glaciers are preconditioned for meltwater ponding and eventually proglacial lake

development, which often result from a deepening and coalescence of supraglacial lakes (Benn et al., 2012; Quincey et al., 2007) which are bounded by a stagnant, ice-cored moraine dam. The combination of the morphology, insulating characteristics of debris and lake development may cause a response to climate forcing that is strongly non-linear (Benn et al., 2012), though only little is known how such a transition develops.

Two key factors can be identified which make lake-terminating glaciers distinctively different from their land-terminating

counterparts, namely the stresses at the bed and the terminus of the glacier. Firstly, a body of water exerts a buoyancy force on the host glacier, reducing the effective pressure and consequently reducing the basal resistance, which ultimately can result in faster glacier flow (Benn et al., 2007b). Secondly, dynamical changes result from processes that act at the terminus and trigger a retreat and reduce along-flow resistive stresses (Nick et al., 2009), which can be especially important in rapidly evolving environments (Benn et al., 2007b). In alpine settings, the transition from a land-terminating glacier to a lake-



terminating glacier could therefore change the dynamic regime of the glacier, something that might be partially decoupled from climate (Benn et al., 2012).

Several recent remote sensing-based studies on glacier surface velocity indicated the divergent evolution of the dynamics of lake- and land-terminating glaciers. Dehecq et al. (2019) documented widespread land-terminating glacier slowdown since the start of the 21ˢᵗ century across High Mountain Asia (HMA) in response to diminished driving stress caused by long-term ice

thinning. In contrast, more localised studies have shown several examples of lake-terminating glacier flow acceleration over a similar time period (King et al., 2018; Liu et al., 2020; Song et al., 2017; Tsutaki et al., 2019). The number and total area of proglacial lakes in the Himalayan region has increased (Nie et al., 2017; Shugar et al., 2020; Zhang et al., 2015), a trend which is likely to continue in the near future, as many glaciers are situated within overdeepenings (Linsbauer et al., 2016). Therefore, a robust understanding of the behaviour of lake-terminating glaciers is crucial. However, a spatially comprehensive assessment

of the contrasting dynamics between land-terminating and lake-terminating glaciers has yet to be undertaken.

Numerical ice-flow models have been utilised to investigate the dynamic thinning of marine-terminating outlet glaciers (Benn et al., 2007a; Enderlin et al., 2013; Nick & Oerlemans, 2006; Vieli et al., 2001; Vieli & Nick, 2011) and more recently of lake-terminating glaciers in alpine regions (Sutherland et al., 2020; Tsutaki et al., 2019). Tsutaki et al. (2019) employed a diagnostic 2-d model setup to show that the transition from a land to a lake-terminating boundary condition will significantly increase the

surface flow velocities near the calving front. Sutherland et al. (2020) used a numerical transient model setup to show that a proglacial lake was a dominant control on the ice velocity during times of glacial lake growth after the Last Glacial Maximum in New Zealand. Ice thickness data and a quantification of the hydrological characteristics of a glaciers proglacial lake are important components of such model setups (Carrivick et al., 2020), but these data are very limited in the Himalayas, making a realistic model setup problematic.

The main aim of this study is to examine the influence of proglacial lakes on Himalayan glacier dynamics, in order to improve the current understanding of the large subregional heterogeneity of glacier behaviour. More specifically, we seek to investigate the attribution of lake-driven changes in the velocity field to dynamic thinning and investigate the role that debris cover plays on glacier-lake dynamics. To do so we used Sentinel-2 satellite imagery to derive a large-scale contemporary Himalayan glacier velocity dataset at an improved resolution compared to studies to date. We compare the velocity dataset against surface

elevation change data (King et al. 2019) to discuss the role of proglacial lakes and debris cover on glacier dynamics. Finally, we employ a numerical flow model to investigate the factors that dominate the lake-terminating dynamics and explore their potential to accelerate current and future mass losses.



## 2 Study Area

Our study area covers five sub-regions within the Central and Eastern (CE) Himalaya (Fig. 1). Glaciers in the CE Himalaya cover an area of ~13,900 km$^2$, which is about 60 % of the glacierised area of the total Himalayan arc (Bolch et al., 2012). The Himalayas are located around the southern rim of the Tibetan Plateau (TP), and the CE Himalayas are the source of two major trans-boundary rivers, namely the Ganges and the Brahmaputra. The extreme Himalayan topography exerts a strong influence on north-south contrasting precipitation patterns by forming an orographic barrier and depleting the monsoonal air of the bulk of its moisture on southern windward slopes, resulting in relatively dry slopes on the TP (Ageta and Higuchi, 1984).

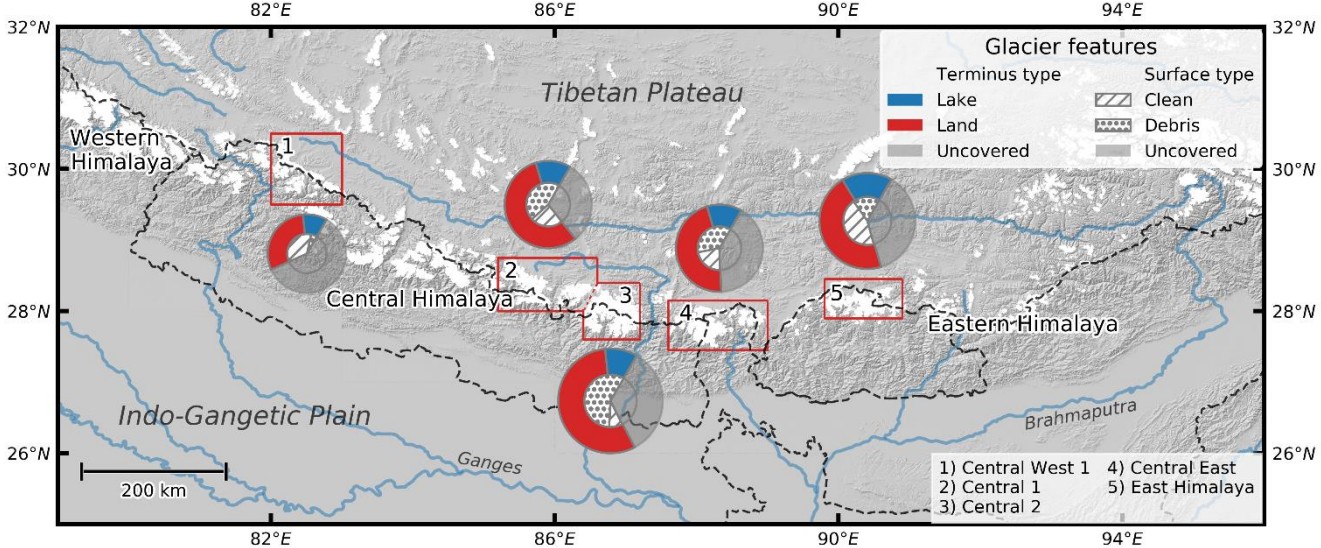

**Figure 1: Map showing the regional subdivisions (red rectangles) with the associated glacier characteristics, including terminus type and surface cover as a fraction of the total sub-regional glacierised area. The uncovered fraction represents the glacierised area from glaciers smaller than 3 km$^2$. The names of the subregions are in accordance with King et al. (2019). Country boundaries are tentative and for orientation only. This figure was generated using Matplotlib 3.1.2, together with Python 3.7.**

Related to this stark contrast in north-south relief is the distribution of clean-ice and debris-covered glaciers (Scherler et al., 2011). Glaciers in low-relief areas sloping northwards facing the TP generally show little or no debris cover and have extensive accumulation areas. In contrast, glaciers surrounded by much steeper topography on the southern side of the main orographic divide receive a large proportion of their accumulation by snow avalanching from steep hillslopes. Steep hillslopes supply large fluxes of rocky material to the glacier and, as a result, glaciers in such settings often have an extensive debris cover, which can range from a few centimetres to several meters in thickness (Scherler et al., 2011).

In this study we only focus on glaciers with an area larger than 3 km$^2$, compared to the 5 km$^2$ previously utilised by Dehecq et al. (2019). For this we used the glacier outlines and corresponding surface area from The Randolph Glacier Inventory (RGI 6.0) (The RGI Consortium, 2017). Glaciers smaller than 3 km$^2$ are often located at a high elevation and do not typically host a proglacial lake, and thus fall beyond the scope of our study. Also, small glaciers often show only very low surface velocities,





and in general glacier volume scales exponentially with glacier surface area, which increases the representativeness of our study onto potential ice volume losses.

To study glacier-lake dynamics, we select five sub-regions within the CE Himalayas as covered by the study of King et al. (2019) with a high density of proglacial lakes, following the lake inventory of Zhang et al. (2015) (Fig.1; Table 1). We classify
glaciers as lake-terminating when the glacier shows a clear terminal ice cliff in direct contact with the lake and base our classification on the glacial lake inventory of Wangchuk & Bolch (2020) and Zhang et al. (2015) using multiple sources of satellite imagery. The classification of debris cover is binary (debris-covered or clean-ice) and for this we follow the criteria defined by Brun et al. (2019), classifying glaciers as debris-covered where more than 19 % of their area was mantled by debris.

**Table 1: Regional distribution of lake-terminating, land-terminating, debris-covered and clean-ice glaciers. *The coverage over the whole region is relatively low since it also incorporates all CE Himalayan glaciers outside the five subregions.**

| | Terminus Type | | Surface Cover | | Area | Coverage |
|---|---|---|---|---|---|---|
| | Number of Glaciers & (% Area) | | | | | |
| **Subregion** | Lake | Land | Debris | Clean | | |
| Central West 1 | 6 (24 %) | 33 (76 %) | 7 (12 %) | 32 (88 %) | 920 km$^2$ | 40 % |
| Central 1 | 16 (18 %) | 53 (82 %) | 36 (66 %) | 33 (34 %) | 1098 km$^2$ | 69 % |
| Central 2 | 10 (15 %) | 57 (85 %) | 44 (87 %) | 23 (13 %) | 1404 km$^2$ | 66 % |
| Cenrtral East | 17 (21 %) | 57 (79 %) | 34 (62 %) | 40 (38 %) | 1130 km$^2$ | 59 % |
| East Himalaya | 20 (27 %) | 49 (73 %) | 19 (29 %) | 50 (71 %) | 1238 km$^2$ | 63 % |
| **All** | 70 (21 %) | 249 (79 %) | 139 (57 %) | 178 (43 %) | 5781 km$^2$ | 41 % * |



# 3 Methods

## 3.1 Surface Ice Velocity

### 3.1.1 Sentinel-2 Satellite Imagery

The European Copernicus Sentinel-2 series consists of two satellites; Sentinel-2a, launched in June 2015 and Sentinel-2b, launched in March 2017. The two satellites have a combined revisit time of 5 days and their orthorectified image products
(Level-1C) are freely available at https://earthexplorer.usgs.gov/ and https://scihub.copernicus.eu/. In this study the 10 m near-infrared band (band 8) was used to exploit the contrasting spectral properties of fresh snow, firn and clean ice at this wavelength, an approach which has proven to work well for feature tracking on a variety of glacier surfaces (Kääb et al., 2016). Throughout most of its mission, the multi-temporal co-registration accuracies of Sentinel-2 products from the same orbit have been below 12 m (95.5 % confidence interval), which is reported monthly by the European Space Agency (ESA). When co-
registering two Level-1C images from the same relative orbit (repeat-orbit), DEM effects will be present but have the same pattern in both datasets (because they have a similar off-nadir cross-track look angle) so that they can be eliminated by calculating the average offset field obtained from correlating the two images. Products acquired from neighbouring overlapping swaths translate into additional offsets of up to 5.9 m (Kääb et al., 2016), and have therefore been omitted in this study.

### 3.1.2 Image Pair Selection

We selected multiple image pairs of the same locality to maximise velocity field coverage, which is often limited by shadows, cloud cover, low visual contrast, or sensor saturation, and to increase the overall confidence in the velocity estimates. The final velocity field is then an average of all the valid velocity estimates, a strategy explored by several studies (i.e., Dehecq et al., 2015; Scherler et al., 2011; Willis et al., 2012). The maximum number of image pairs separated by one year was selected for the month of November, as this month is associated with low cloud cover and a relatively high snow line. Sourcing imagery
from the same season often results in similar surface conditions and consequently improves the image matching algorithm performance. This approach produced a dataset of 149 images and 427 image-pairs with an effective date at 24 August 2018 (Table 2). Note that due to the lower Sentinel-2 repeat cycle in 2016, this effective date is centred towards the end of the November 2016 - November 2019 interval.

### 3.1.3 Preprocessing

To reduce computational costs, a mask was applied over all non-glacierised areas and glaciers with an area below 3 km$^2$. For this mask we used the glacier outlines from The Randolph Glacier Inventory (RGI 6.0) (The RGI Consortium, 2017). We selected an area off-glacier of about 300 km$^2$ in each tile, where the displacement is expected to be zero, to evaluate the precision and uncertainty of the feature tracking algorithm and to reduce the co-registration error.

Heid & Kääb (2012) evaluated several feature tracking methods and showed that orientation correlation performed best under
most circumstances. This method, developed by Fitch et al. (2002), creates two complex orientation images from the original



image pairs and was adopted by this study. Each pixel represents the orientation of intensity gradient at that pixel, making the method invariant to illumination change, which is a desired property for feature tracking algorithms.

**Table 2: November Sentinel-2 images, image pairs and effective date between 2016 – 2019.**

| Satellite Tile | Number of Images | Number of Pairs | Effective Date |
|---|---|---|---|
| T44RPU | 15 | 46 | 16–10–2018 |
| T44RPT | 16 | 58 | 28–10–2018 |
| T45RTN | 14 | 39 | 07–07–2018 |
| T45RTM | 13 | 34 | 25–05–2018 |
| T45RUM | 12 | 26 | 08–07–2018 |
| T45RVM | 19 | 76 | 20–09–2018 |
| T45RVL | 12 | 29 | 31–05–2018 |
| T45RWL | 11 | 25 | 01–07–2018 |
| T45RXL | 11 | 22 | 23–09–2018 |
| T45RYM | 13 | 34 | 09–10–2018 |
| T46RBS | 13 | 38 | 09–12–2018 |
| **Total** | 149 | 427 | 24–08–2018 |


### 3.1.4 Image Matching

From the two orientation images a search window (i.e., a squared collection of pixels) $f_c(i, j)$ and a reference window $g_c(i, j)$ centred around the same location (x, y) were extracted and matched using correlation computed in the frequency domain with Fast Fourier Transforms (FFTs), according to the convolution theorem (McClellan et al., 1999). Given $F_C(k, l)$, the FFT of

$f_c(i, j)$, $G_C^*(k, l)$, the complex conjugate of the FFT of $g_c(i, j)$, and $\Re\{IFFT()\}$ the real part of the Inverse Fast Fourier Transform function, the orientation correlation $CC(m, n)$ matching surface is:

$$CC(m, n) = \Re\{IFFT(F_C(k, l)G_C^*(k, l))\}, \tag{1}$$

The registration of $f_c(i, j)$ and $g_c(i, j)$ was measured from the position of the maximum in $CC(m, n)$. We matched the orientation of the intensity gradient that is contained in the phase of the orientation image. After this initial estimate we then

refined the maximum estimation by upsampling the product $F_C(k, l)G_C(k, l)$ in a small neighbourhood of the initial maximum (Guizar-Sicairos et al., 2008). This matching process was repeated over all the glacierised areas with steps equalling half of the search window size, for all image pairs. This leaves us with n-pairs of velocity data matrices (x- and y-displacement) with a resolution of half times the search window size for each given satellite scene (Table 3).





**Table 3: Parameters used in image matching using Sentinel-2 10 m pixel resolution imagery.**

| Image Matching Parameters | Number of Pixels | Size |
|---|---|---|
| Search window size | $16 \times 16$ | $160 \text{ m} \times 160 \text{ m}$ |
| Reference window size | $46 \times 46$ | $460 \text{ m} \times 460 \text{ m}$ |
| Search limit | 23 | 230 m |
| Iteration step | 8 | 80 m |
| Subpixel resolution | 1/16 | 0.625 m |

### 3.1.5 Postprocessing

To remove matching blunders present in the derived velocity fields we largely adopted a strategy proposed by Gardner et al. (2020). We used a disparity filter with two components. First, the filter checks for the `uniqueness' for each component velocity

by comparing each element with their surrounding neighbours that are co-located in a 5 by 5 kernel. If less than 9 of the 25 co-located are within a 25 % range of the search limit of the algorithm, the velocity component is too unique and is disregarded. For each 100 by 100 km tile, we selected a large stable area from which to calculate the median offset in x- and y-direction. To reduce the noise in the velocity data, we subtracted this median offset from the whole (glacierised and stable) x- and y-displacement field. The 2017-2019 final x- and y-displacement field was then created by taking the median of all the image

pairs for both velocity components (Pronk et al., 2021).

### 3.2 Uncertainty of the Velocity Field

Uncertainties of the final median velocity field are dominated by the precision of the feature-tracking algorithm, the coregistration error, the temporal variability of glacier flow and the number of velocity estimates. For the estimation of the 95 % confidence interval ($CI_{95}$) of each median velocity component, we adopted the methodology of Dehecq et al. (2015), and

expect the $CI_{95}$ to conform to:

$$CI_{95} = \kappa \frac{MAD_{disp}}{N^{\alpha}}, \tag{2}$$

where $MAD_{disp}$ is the dispersion at each velocity location of the N number of estimates:

$$MAD_{disp}(i, j) = 1.483 \times median_{t \in T}\{|V(i, j, t) - \overline{V}(i, j)|\}, \tag{3}$$

where T is the collection of N velocity estimates $V(i, j, t)$ merged to obtain the median velocity $\overline{V}(i, j)$ at pixel (i,j). Parameters

$\kappa$ and $\alpha$ determine the width and the thickness of the tail of the distribution and have yet to be estimated. Equation (2) leaves



us with three unknowns but can be solved at stable areas where $CI_{95}$ can be determined as a function of $MAD_{disp}$ and N for each tile location with its corresponding stable area, providing an uncertainty estimation for areas with actively flowing ice.

**3.3 Surface Elevation Change, Estimation of ELA and Surface Slope**

We examined ice thinning rates using the surface elevation change (dh dt$^{-1}$) dataset from King et al. (2019). The elevation
change field is a mean estimate derived from 499 DEMs generated from WorldView and GeoEye optical stereo pairs spanning the period 2012-2016, with an effective date around mid-2015 (Shean, 2017), and the Shuttle Radar Topographic Mission (SRTM) DEM from 2000. The effective dates of our elevation change and surface velocity datasets are therefore separated by ~3 years. Considering an average retreat rate of $26.8 \pm 1.4$ m yr$^{-1}$ for lake-terminating glaciers (King et al. 2019) the lowermost 100 m of lake-terminating glaciers are likely to be devoid of surface velocity data when the two datasets are compared.
However, this could be considerably more for fast retreating glaciers, which must be considered when interpreting our results. In this study we focused our analyses on the ablation zone of glaciers, for which we need an estimation of the equilibrium-line altitude (ELA). We followed the approach of Braithwaite & Raper (2009) and considered the median altitude of each glacier as a proxy for the ELA, defined by the elevation of the 50$^{th}$ percentile in glacier area. This estimate of the ELA is available within the Randolph Glacier Inventory (RGI- 6.0) by the RGI Consortium (2017) (Pfeffer et al., 2014).
To calculate the slope of the ablation zone we used the Advanced Land Observing Satellite (ALOS) World 3D DEM (Tadono et al., 2014), which is available at a 30 m resolution and is based on DEMs generated from stereo image pairs collected over the period 2006-2011.

**3.4 Glacier Centre Flow Line Analysis**

We analysed surface velocity, elevation change and slope along the main glacier centre flow line (hereafter, centreline), an
approach adopted by several earlier studies (i.e., Liu et al., 2020; Nagler et al., 2015; Scherler et al., 2011). Centrelines were produced with the Open Global Glacier Model (OGGM) (Maussion et al., 2019) using a slightly adapted algorithm from Kienholz et al. (2014), and glacier outlines from RGI 6.0 using the SRTM DEM. All centrelines were manually adapted using 2019 Sentinel-2 satellite data and velocity data from this study to ensure that the centrelines end at the 2019 terminus position and that they follow the main flow tributary.
To extract centreline velocity data, we conducted a nearest neighbour sampling every 80 meters, and averaged the velocity estimate by using a 3 by 3 (240 m by 240 m) Gaussian window:

$$u_c = \sum_{j,i=-1}^{1} \frac{u_{i,j}}{\left(CI_{95_{i,j}}\right)^2} e^{-\frac{1}{2}\frac{i^2+j^2}{\sigma^2}}, \tag{4}$$

where u is the velocity estimate at pixel (i,j), $CI_{95}^{-2}$ is the weighting factor and where $\sigma = 0.7$ is the standard deviation of the Gaussian window. This approach increases the overall confidence of our median velocity estimates. The Gaussian window





also prevents pixels further away from the centre flow line skewing the averaged data, which may result in an underestimation of the velocity values.

Then, to compare the velocity profiles for multiple glaciers at the ablation zone, we normalised the glacier centrelines horizontally along their ablation zone length. To achieve this, we first selected all discrete centreline velocity data points starting at the ELA, and upsampled all centrelines with an ablation area length below 4000 m and downsampled the rest. We

took 4000 m as the most representative length as the results showed that this approaches the overall median length of the ablation zone for the whole glacier population. Our choice to analyse the centreline data along the normalised length of the ablation zone provides information on the dynamic influence of terminus type and surface cover but limits our ability to evaluate the effect of climate on surface elevation change, such as done by King et al. (2019) and Maurer et al. (2019). This also restricts the possibility to quantitively attribute contrasting surface elevation change rates of lake-terminating and land-

termination glaciers to dynamic thinning, which is especially true for clean-ice glaciers whose thinning rates appear to be highly dependent on climate, hence elevation (Scherler et al., 2011). Therefore, this study will be restricted to a qualitative analysis when evaluating the velocity data in the context of surface elevation change rates.

**3.5 Glacier Group Uncertainty**

The glacier group uncertainty (i.e., the uncertainty in the median velocity of a collection of glaciers along the normalised

flowline) depends on the uncertainty of the individual glacier velocity measurements ($CI_{95}$), the spread between the velocity points $u_c$ among the sample group and the number of velocity points $N_u$. We estimated this uncertainty by applying a Monte Carlo simulation which draws 200 random points from the uncertainty distribution of each individual velocity point $u_c$ in the region of interest. Then for each sample round, following the bootstrap method, we drew $N_u$ samples with replacement to calculate the median, and repeated this 500 times. This results in $10^5$ estimates of the median from which we determined one

standard error (SE) interval, and we used this as primary estimator of our regional mean median velocity uncertainty.

**3.6 Numerical Flowline Model**

**3.6.1 Model Description**

To investigate the processes which may be influencing the velocity regime of lake-terminating glaciers in alpine settings, we carried out a synthetic diagnostic numerical experiment by using a flowline model that was previously utilised on

predominantly marine terminating outlet glaciers (Enderlin et al., 2013; Nick et al., 2010; Nick et al., 2009; Vieli & Payne, 2005; Vieli & Nick, 2011). The depth integration of the model implicitly employs the Shallow Shelf Approximation (SSA), which is not fully appropriate for the entire model domain. However, the model results in the ablation zone, where the surface slope is generally low and where we assume sliding to dominate the glacier flow (Liu et al., 2020; Tsutaki et al., 2019), should still be adequate to provide us useful insight in the relevant physical processes in operation (Le Meur et al., 2004). The

governing force balance equation determined through conservation of momentum is (Nick et al., 2010):




$$2\frac{\partial}{\partial x}\left(Hv\frac{\partial U}{\partial x}\right) + A_s\left[\left(H - \frac{\rho_w}{\rho_i}D\right)U\right]^{\frac{1}{m}} + \frac{H}{W}\left(\frac{5U}{2AW}\right)^{\frac{1}{3}} = \rho_i gH\frac{\partial h}{\partial x},\tag{5}$$

where $U$ is the vertically averaged horizontal ice velocity, $\rho_i$= 917 kg m$^{-3}$ is the density of ice, $\rho_w$= 1000 kg m$^{-3}$ is the density of fresh water, $H$ is the ice thickness, $A_s$ is the sliding parameter, $D$ is the ice thickness submerged under the lake level, $W$ is the glacier width, $g$ is gravitational acceleration, $h$ is the ice surface elevation, $m$ is the bed friction exponent, and $v$ is the depth averaged effective viscosity, which is defined as follows:

$$v = A^{-1/3}\left|\frac{\partial U}{\partial x}\right|^{-\frac{2}{3}}\tag{6}$$

The right-hand side of Eq. (5) is the gravitational driving stress, which is balanced by gradients in longitudinal stress (1st term left hand side), basal resistance (2nd term), and lateral resistance (3rd term). $A$ is the temperature-dependent rate factor and increases from a minimum of $3.5 \times 10^{-25}$ Pa$^{-3}$ s$^{-1}$ at the divide to a maximum of $1.7 \times 10-24$ Pa$^{-3}$ s$^{-1}$ at the calving front, corresponding to a depth-averaged ice temperature range of $-10°$ C to $-2°$ C at the ablation zone (Cuffey and Paterson, 2010), for which we follow Enderlin et al. (2013).

The assumption is made that basal drag depends on sliding velocity and effective basal pressure nonlinearly (Bindschadler, 1983; van der Veen & Whillans, 1996; Vieli & Payne, 2005), for which we choose m = 3. Resistance from drag along the lateral margins is estimated by integrating the force-balance equation over the width of the glacier assuming a constant ice thickness that lateral drag supports the same fraction of driving stress along a transect across the glacier. Consequently, the model assumes a flow through a rectangular basin, with lateral support that is independent from effective pressure. The up-glacier boundary is the upper bound of the glacier (U = 0) and at the calving front the longitudinal stress is balanced with the difference in hydrostatic pressure between the ice and lake water, which results in the following depth-averaged stretching rate:

$$\frac{\partial U}{\partial x} = A\left[\frac{\rho_i g}{4}\left(H - \frac{\rho_w}{\rho_i}\frac{D^2}{H}\right)\right]^3.\tag{7}$$

The synthetic model domain extends 8000 m horizontally, 1000 m vertically and resembles the main characteristics of relatively large clean-ice lake-terminating glacier of which numerous examples can be found flowing northwards onto the TP (Fig. A1, in the appendix). We assumed a concave-up profile resulting in a slope of about 4.5° within 2 km of the terminus, and our interpretation in the results will solely be focused on this part of the glacier. For glacier width (W), we used a value of 600 m at the terminus which increases to 2.5 km up-glacier to reduce the influence of lateral margins. We used a maximum ice thickness (H) of 230 m and an ice thickness of 120 m at the terminus, values in line with ice-thickness estimates of the larger Himalayan glaciers (Farinotti et al., 2019). Subglacial water pressure is assumed to follow a piezometric surface rising up-glacier (Benn et al., 2007b), and we estimated the piezometric surface to be equivalent to 60 % of the ice thickness away from the calving front, accounting for a simplified basal hydrology. We then tuned the sliding parameter ($A_s$) such that the maximum up-glacier velocity reaches a typical value of 50 m yr$^{-1}$ and found a value of $A_s = 2.5 \times 10^6$ Pa m$^{-2/3}$ s$^{1/3}$.





### 3.6.2 Experimental Design

To examine the importance of the frontal boundary condition, we varied the height of the terminal ice cliff above buoyancy by lowering the lake level by

$$D = \frac{\rho_i}{\rho_w} H_t + \Delta D, \tag{8}$$

where $D$ was the ice thickness submerged under the lake level (i.e., the lake depth at the terminus), $H_t$ is the ice thickness at the terminus and $\Delta D$ is the lake surface level change. We examined the glacier dynamics when the calving front is exactly buoyant (i.e., $\Delta D = 0m$) and for an increased ice cliff height resulting from a lake surface level change of -10 m and -15 m, keeping the terminus position and ice thickness constant. The varying height of the terminal ice cliff above buoyancy are chosen to fall within a realistic range based on the limited observational evidence on terminal ice cliffs (Watson et al., 2020).

For each ice cliff height (i.e., lake surface level) configuration, we also ran the numerical model by keeping the basal friction independent from the effective pressure $(H - \frac{\rho_w}{\rho_i} D = 1)$, ruling out the effect of the lake on basal friction, and used a corresponding roughness parameter of $A_s = 9 \times 10^6$ Pa m$^{-1/3}$ s$^{1/3}$. Note that lowering the lake surface is just one way of manipulating the frontal boundary condition, and that an instantaneous glacier retreat could be an alternative way to alter this balance. Also note that this experimental design investigates lake-terminating glaciers under varying frontal and basal

conditions and prohibits the analysis of a land-terminating glacier, which would ask for a different glacier terminus morphology.

We performed a basic sensitivity analysis in the situation of $\Delta D = -10\ m$ by varying the glacier width by $\pm 300$ m, ice thickness by $\pm 50$ m and surface slope by $\pm 1.5°$. We performed additional analyses to investigate the model sensitivity to the large current uncertainty of ice-thickness estimates for Himalayan glaciers (Farinotti et al., 2019) by varying the ice thickness

again but keeping the maximum velocity at 50 m yr$^{-1}$ and tuning $A_s$ accordingly.



# 4 Results

## 4.1 Algorithm Performance and Velocity Accuracy

The coregistration error calculated on non-glacierised, stable areas enabled us to reduce the dispersion ($MAD_{disp}$) by 56 %, resulting in a $MAD_{disp}$ distribution with a median at 4.15 m yr$^{-1}$ (Fig. 2a). The distribution is heavy-tailed, with the largest uncertainties found over accumulation zones where the algorithm was unable to remove all mismatches (Fig. A2, in the appendix). Another large source of uncertainty is the interannual variability in glacier flow, resulting in high dispersion in areas with an overall high flow velocity. The $CI_{95}$ distribution is slightly less heavy-tailed than $MAD_{disp}$, with a median uncertainty just below 3 m (Fig. 2b).

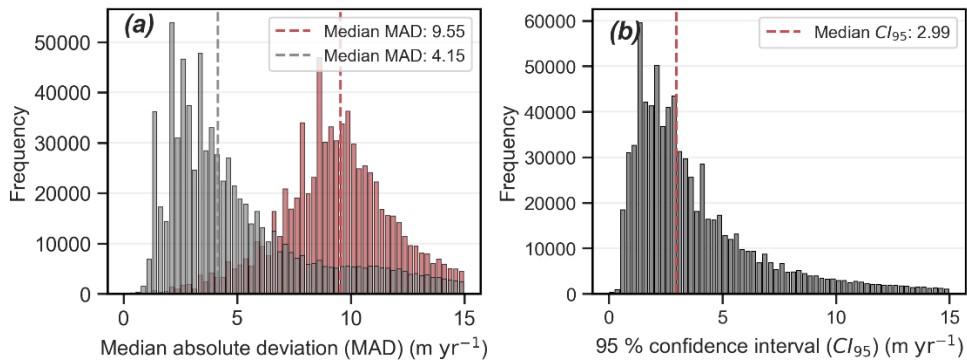

**Figure 2: Dispersion ($MAD_{disp}$) (a) and 95 % confidence interval ($CI_{95}$) (b) of the velocity estimates at glacierised areas. (a) The red distribution represents the $MAD_{disp}$ before subtracting $V_{off}$ from each image pair, which realised a 56% reduction of the median dispersion, resulting in the grey dispersion distribution. (b) $CI_{95}$ resulting from the estimated distribution of median displacement vector as a function of the number of velocity estimates and $MAD_{disp}$ (Eq. 2).**

When evaluating the $CI_{95}$ along the centrelines, a consistent trend is apparent (Fig. 3). The uncertainty decreases from the ELA moving further into the ablation zone due to the enhanced pixel contrast. Close to the terminus however, the uncertainty rises again, which can be related to relatively large interannual changes in surface properties, resulting in reduced algorithm performance. Interestingly, lake-terminating glaciers have consistently higher uncertainty along the ablation zone, which likely results from the large velocity differences between lake-terminating and land-terminating glaciers (Section 4.3). In the following sections we consider velocity estimates with a $CI_{95}$ larger than 5 m yr$^{-1}$ too uncertain, and these estimates are removed from further analyses.

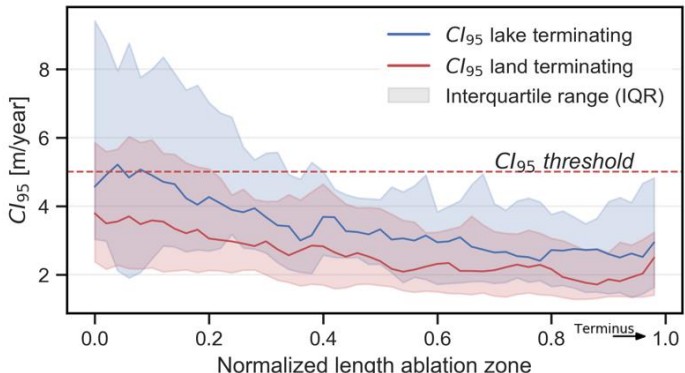

**Figure 3: Median CI$_{95}$ (m yr$^{-1}$) of the velocity estimates for lake-terminating and land-terminating glaciers along the normalised glacier centre flow line at the ablation zone, with the terminus positioned at the right end of the figure. The spread among the glacier population is represented by the interquartile range (IQR). Velocity estimates above the CI$_{95}$ threshold (5 m yr$^{-1}$) are removed from**
**the dataset.**

## 4.2 Comparison to Other Glacier Velocity Datasets

The lack of ground-truth velocity measurements hinders simple evaluation of remotely sensed measurements in most cases (Scherler et al., 2008). To assess the quality of our measurements we compare them with two region-wide velocity datasets, both processed with predominantly optical Landsat-8 imagery. Dehecq et al. (2019a) produced a composite glacier surface

velocity field for the Pamir-Karakoram-Himalaya for the years 2013-2015. Velocity fields are available at 120 m resolution and produced using a 240 m reference window (Dehecq et al., 2015). Another region-wide dataset is generated using auto-RIFT (Gardner et al., 2020) and provided by the NASA MEaSUREs ITS_LIVE project (Gardner et al., 2020). This velocity field spans from 1985 to 2020, but we compare our data to an ITS_LIVE velocity field with an effective date of around spring 2018, which is available a 120 m resolution and again is computed using a 240 m reference window.

Substantial differences exist in the region-wide median centreline surface velocity (later referred to as velocity) between the three datasets, with maxima ranging from just above 5 m yr$^{-1}$ (Gardner et al., 2019) to well above 13 m yr$^{-1}$ in our study (Fig. 4a). Velocity measurements from the three datasets agree reasonably well close to the terminus (within 0.5 m yr$^{-1}$ range) where velocities are expected to be close to stagnant, which indicates that differences between flow fields are proportional to the magnitude of the regional median centre line velocity. Dehecq et al. (2019) observed a slowdown of glacier velocities for all

our subregions, ranging from -14.5 ± 1.3 % to -21.0 ± 2.3 % per decade, in response to climate induced changes in slope and ice thickness. These reductions in surface velocity only partly explain the differences in velocity between Dehecq et al. (2019a) and Gardner et al. (2019).

The limited width of some Himalayan valley glaciers, which can be as little as 300 m, may explain discrepancies in the velocity fields derived using different methods. Narrow valley glaciers are subject to considerable lateral stress induced transverse

velocity gradients. A large reference window size (g$_c$) might consequently result in an underestimation of the centreline velocity as it is simply unable to resolve this velocity gradient. The usage of a variable reference window size by Dehecq et al.



(2019a) and Gardner et al. (2020) up to 4 times the original reference window size could also potentially explain these differences, although we are unable to verify this as the spatial variability of the effective reference window size is not documented in these datasets. Selecting only large glaciers, often with a wider ablation zone, largely diminishes this discrepancy (Fig. 4b), which supports our hypothesis, indicating that the employment of the Sentinel-2 satellites improved the resolution and therefore the analytical potential of the glacier centreline velocity data.

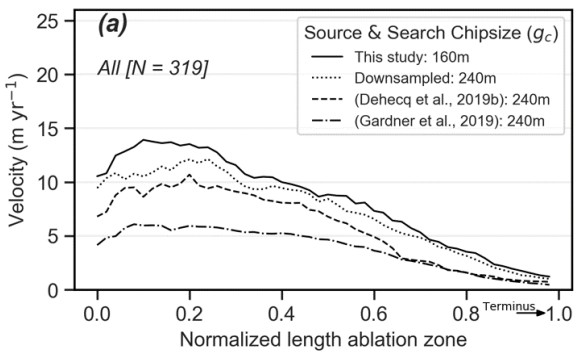 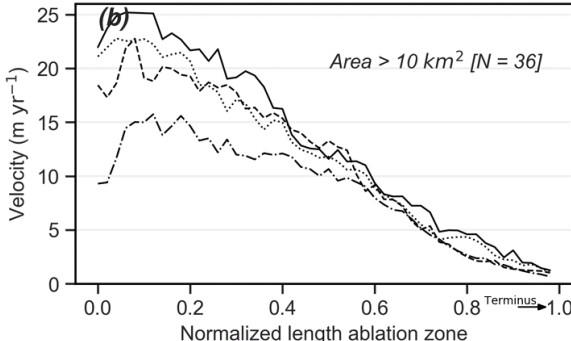

**Figure 4: Regional median centre flow line surface velocity (m yr⁻¹) comparison between Dehecq et al. (2019b), Gardner et al. (2019) and this study of glaciers greater than 3 km² in area (a) and glacier greater than 10 km² in area (b).**

**4.3 Terminus Type Variability in Velocity**

Our velocity analysis shows (Fig. 5a) that the along-flowline mean of median centreline surface velocities (later referred to as mean velocity) of lake-terminating glaciers ($18.8 \pm 0.41$ m yr⁻¹) is substantially higher than the mean velocity of land-terminating glaciers ($8.24 \pm 0.12$ m yr⁻¹) (Table 4). Differences are negligible at the ELA but become steadily larger throughout the ablation zone. Over the lower ablation zone, the differences in surface velocity reach 13.8 m yr⁻¹ (mean velocity of $17.7 \pm 0.47$ m yr⁻¹ for lake-terminating and $3.9 \pm 0.09$ m yr⁻¹ for land-terminating glaciers) (Table A1). Land-terminating glaciers show a stagnant terminus with only little spread around the median velocity among the glacier population. On the contrary, the median velocity of lake-terminating glaciers decreases only slightly, but show a very large spread, indicating a large heterogeneity in lake-terminating dynamical behaviour.

Overall, lake-terminating glaciers cover a larger surface area and show a slightly higher mean surface slope over the ablation zone (Table 5), which might partially contribute to the overall contrast in the mean velocities (Bahr et al., 1997; Scherler et al., 2011). However, this does not explain the large contrast in velocity or heterogeneity at the glacier terminus. Interestingly, when only focusing on the lowermost portion of the ablation zone, where lake-land terminating velocity contrast is greatest, the mean surface slope of lake-terminating glaciers ($-7.2 \pm 3.7$ °) is within the range of the slope of land-terminating glaciers ($-8.2 \pm 4.54$ °) (Table A1), suggesting that factors other than slope are responsible for the velocity contrast close to the glacier terminus.





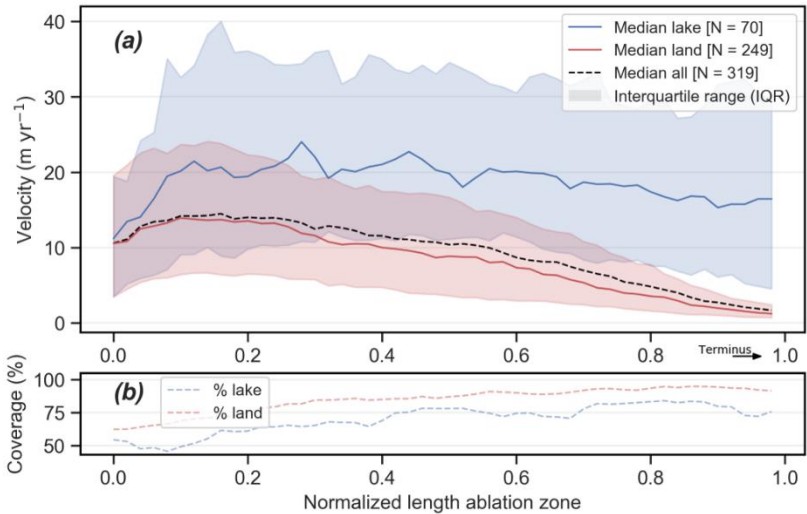

**Figure 5: Median centre flow line surface velocity (m yr⁻¹) (a) and coverage (%) of the velocity estimates (b). (a) The spread among the glacier population is represented by the IQR. (b) The coverage is defined by the percentage of valid velocity estimates (CI₉₅ < 5 m yr⁻¹) at a given position along the centre flowline.**

**Table 4: Mean of median regional centre flow line velocities of lake-terminating and land-terminating glaciers. Uncertainty estimates represent the 1 standard error of the mean (SEM) confidence interval. For the location of the subregions see Fig. 1.**

| | Terminus Type | | |
| --- | --- | --- | --- |
| **Subregion** | **Lake Mean (m yr⁻¹)** | **Land Mean (m yr⁻¹)** | **Both Mean (m yr⁻¹)** |
| Central West 1 | $20.0 \pm 1.57$ | $13.1 \pm 0.40$ | $13.0 \pm 0.40$ |
| Central 1 | $18.3 \pm 0.51$ | $5.44 \pm 0.17$ | $6.72 \pm 0.18$ |
| Central 2 | $11.8 \pm 2.49$ | $6.03 \pm 0.23$ | $6.56 \pm 0.21$ |
| Central East | $18.2 \pm 1.47$ | $8.89 \pm 0.34$ | $10.2 \pm 0.32$ |
| East Himalaya | $27.7 \pm 2.80$ | $10.6 \pm 0.44$ | $13.1 \pm 0.42$ |
| **All** | $18.8 \pm 0.41$ | $8.24 \pm 0.12$ | $9.39 \pm 0.12$ |

**Table 5: Key characteristics of lake-terminating and land-terminating glaciers. Uncertainty estimates represent the 1 standard error (SE) confidence interval.**

| | Terminus Type | | |
| --- | --- | --- | --- |
| **Glacier Feature** | **Lake Mean** | **Land Mean** | **Both Mean** |
| ELA (m.a.s.l.) | $5750 \pm 274$ | $5630 \pm 396$ | $5670 \pm 373$ |
| Area (km²) | $7.48 \pm 4.92$ | $6.40 \pm 4.11$ | $6.68 \pm 4.42$ |
| Ablation Length (m) | $3720 \pm 1602$ | $3920 \pm 2017$ | $3840 \pm 2017$ |
| Slope (degrees) | $-8.8 \pm 41$ | $-8.5 \pm 4.1$ | $-8.6 \pm 4.1$ |





### 4.4 Velocity Dependence on Orientation and Surface Area

We noted that glaciers flowing north onto the TP typically have larger accumulation zones and less debris cover compared to glaciers located in catchments draining to the south of the main Himalayan orographic divide, as also reported elsewhere
(Scherler et al., 2011). Visual inspection indicated that highest velocities are found at such localities, especially in Central West 1, Central 1 and East Himalaya, implying a positive correlation between glacier orientation and mean velocity. Concurrently, a large fraction of the total number of lake-terminating glaciers are orientated northwards, which might falsify the apparent relationship between surface velocity and terminus type proposed in previous section.

To investigate the link between the dynamics and orientation we therefore subdivided our dataset dependent on the orientation
of the glacier ablation zone (Fig. 6a). The results show a large heterogeneity for lake-terminating glaciers, with highest velocities shown for glaciers with their ablation zone orientated to the north. Notwithstanding, for all orientations, lake-terminating glaciers show a higher mean velocity than land-terminating glaciers, although the contrast is minor for glaciers flowing east- or southwards. When only considering the lower half of the ablation zone however, the lake-land terminating velocity contrast becomes substantial for all orientations (Fig. A3a, in the appendix).


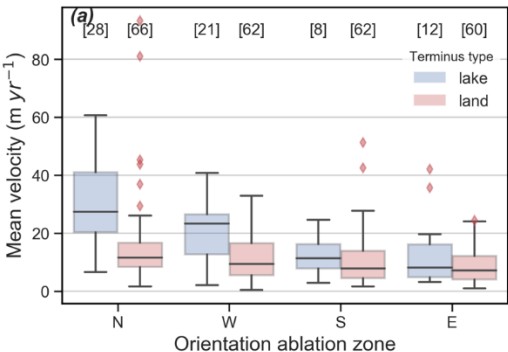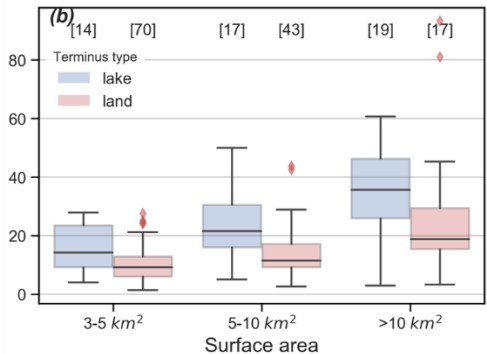

**Figure 6: Boxplot showing the mean velocity contrast between lake-terminating and land-terminating glaciers depending on the orientation of the ablation zone (a) and surface area (b). Boxes represent the IQR of the distribution. Points outside of the 3rd quartile plus 1.5 times the IQR range are plotted explicitly.**

We examined the relationship between glacier surface area and mean glacier velocity by separating glaciers into three area bins with equal sample sizes. Higher mean velocities are apparent for lake-terminating glaciers for each glacier area bin (Fig. 6b), with the largest contrast for glaciers greater than 10 km$^2$. Note that in the largest size bin glaciers are not bounded by an upper area limit, but nevertheless show a comparable median area of $19.2 \pm 8.84$ km$^2$ for lake-terminating glaciers and $18.70 \pm 10.52$ km$^2$ for land-terminating glaciers. Velocity outliers are particularly abundant at large (>10 km$^2$) northward flowing
land-terminating glaciers, such as the clean-ice Zeng Glacier (28°14' N, 90°14' E) in East Himalaya which shows a mean velocity of about 93 m yr$^{-1}$. Again, contrasting velocities between lake-terminating and land-terminating glaciers increase when solely considering the lowermost portion of the ablation zone (Fig. A3b, in the appendix), indicating that regardless of





orientation and size, substantial contrast in glacier surface velocity is related to terminus type, and increases towards the glacier tongue.

**4.5 Regional Variability**

We find large variability in mean velocities between different regions (Fig. 7), with highest mean velocities in Central West 1 (13.0 $\pm$ 0.40 m yr$^{-1}$) and East Himalaya (13.1 $\pm$ 0.42 m yr$^{-1}$) (Table 4), areas with the largest proportions of clean ice. All regions show higher mean velocities for lake-terminating glaciers than for land-terminating glaciers, though large variability between regions is apparent. In Central 2 mean velocity differences between lake-terminating (11.8 $\pm$2.49 m yr$^{-1}$) and land-

terminating glaciers (6.03 $\pm$0.23 m yr$^{-1}$) are relatively modest and coincide with a high proportion of debris-covered glaciers for both terminus types. For Central 1 and East Himalaya, a substantial part of the velocity contrast can be attributed to the relatively high abundance of lakes at large clean-ice northward flowing glaciers, explaining the large velocity contrast which is already substantial at the ELA. Finally, in the regions Central West 1 and Central 1 we observe an increase in velocity towards the terminus, indicating that most of the glaciers accelerate towards the ice-water interface. Trends in velocity of lake-

terminating glaciers in these regions should the treated with caution however, as the population of lake-terminating glaciers is very limited (N = 6, 10). Nonetheless, all regions show a large contrast in heterogeneity close to the terminus, suggesting that the influence of proglacial lakes on glacier dynamics is a region-wide phenomenon.

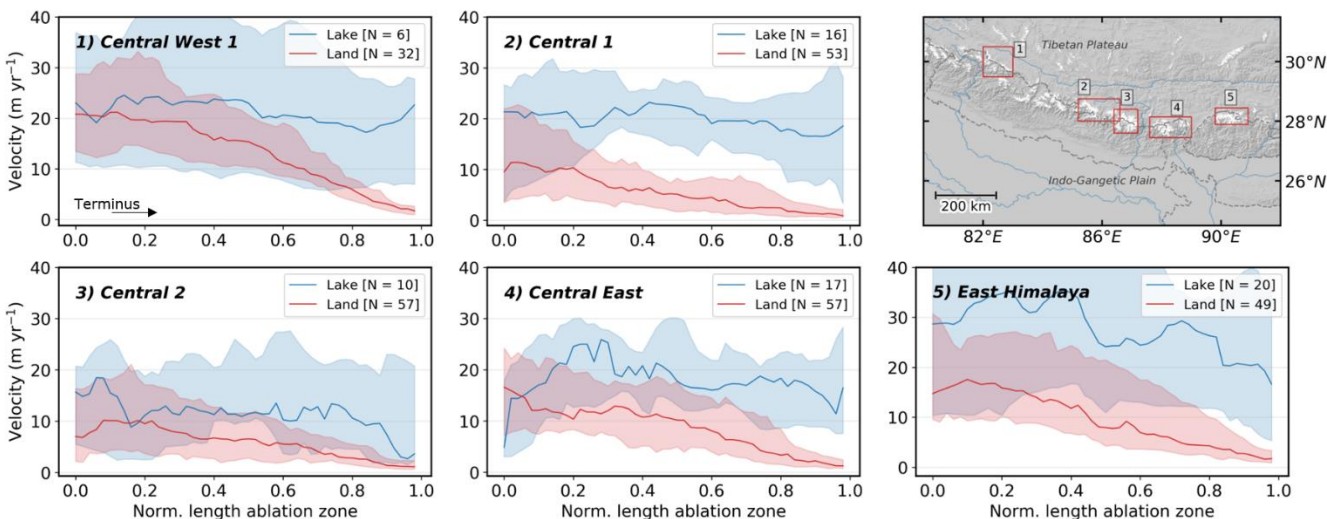

**Figure 7: Subregional glacier median centre flow line velocity estimates and their location along the CE Himalaya (red rectangles). The spread shows the IQR among the glacier population, with the number of glaciers shown in the legend between the brackets. This figure was generated using Matplotlib 3.1.2, together with Python 3.7.**





**4.6 Impact of Surface Cover on Glacier-Lake Dynamics**

To examine the role of debris cover on glacier-lake dynamics, we subdivided our dataset into glaciers with >19% debris cover

and those with < 19% debris cover, which we classify as clean-ice glaciers (Fig. 8a, c, e). We measured substantially higher velocities for lake-terminating glaciers, both debris covered and clean ice (Fig. 8c, e; Table 6), although large differences are apparent depending on surface type. Most notably, the absolute lake-land mean velocity contrast of debris-covered glaciers ($11.5 \pm 0.63$ m yr$^{-1}$ vs $5.96 \pm 0.11$ m yr$^{-1}$) is lower than for clean-ice glaciers ($22.5 \pm 0.52$ m yr$^{-1}$ vs $10.3 \pm 0.14$ m yr$^{-1}$) but indicate for both surface types a doubling in surface velocity when a lake is present.

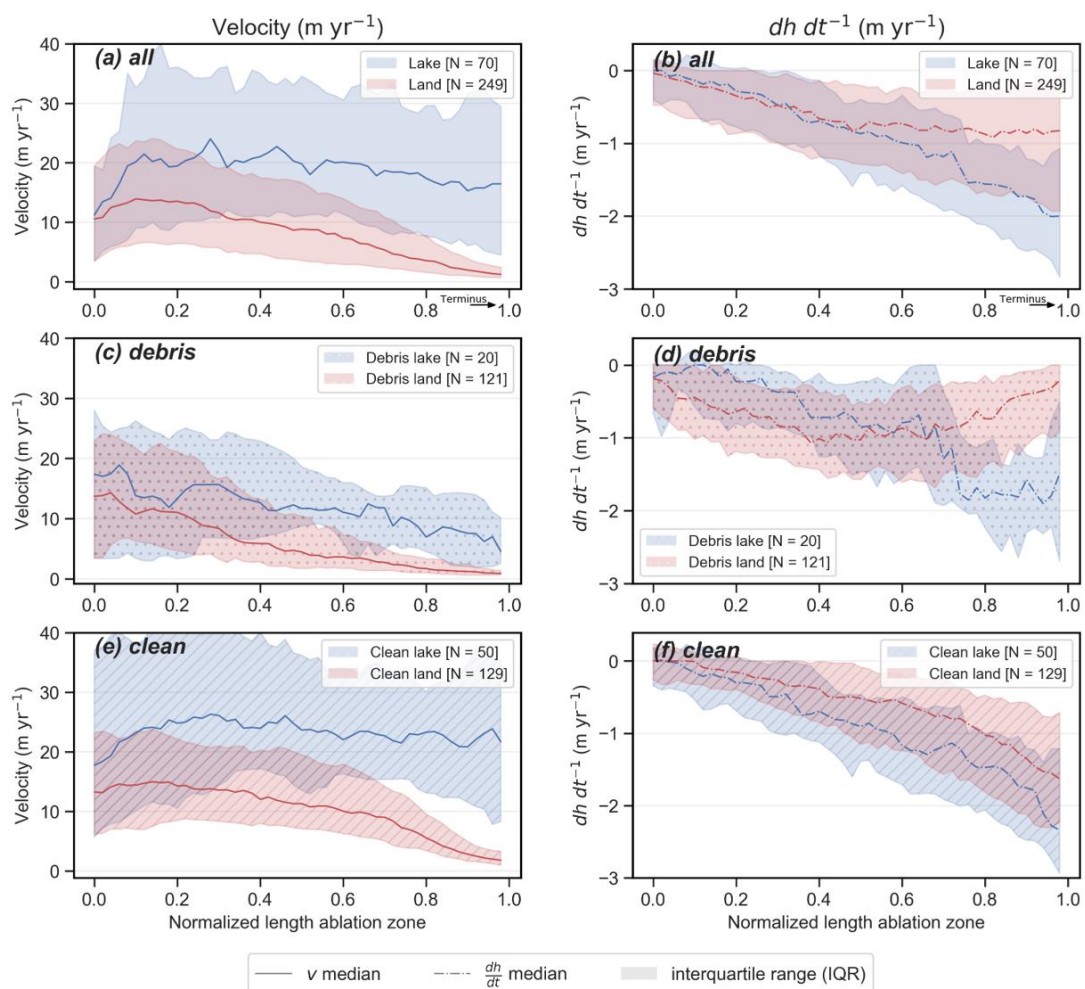


**Figure 8: Glacier median centre flow line velocity (m yr$^{-1}$) (a, c, e) and surface elevation change (dh dt$^{-1}$) estimates (after King et al. (2019)) (b, d, e) for lake-terminating land-terminating glaciers. A further subdivision is made between debris-covered (c, d) and clean-ice glaciers (e, f). The spread shows the IQR among the glacier population.**

For both surface types, higher velocities for lake-terminating glaciers are coincident with elevated surface lowering over

coincident portions of glacier ablation zones (Fig. 8b, d, f). Again, large differences exist depending on surface type, with a





very large contrast in surface lowering close to the termini of lake and land-terminating debris-covered glaciers (about 1.5 m yr$^{-1}$) and less pronounced contrast for clean-ice glaciers (0.5 m yr$^{-1}$). We relate this to the distinct differences in surface mass balance properties between clean-ice and debris-covered glaciers, which becomes clearly visible for land-terminating glaciers. For lake-terminating clean-ice glaciers, the velocity profile remains close to constant towards the terminus and shows a very

large spread among the glacier population (Fig. 8e). Enhanced surface lowering at lake-terminating clean-ice glaciers steadily grows to -0.5 m yr$^{-1}$ towards the terminus and is less pronounced than the surface lowering contrast at debris-covered glaciers. We find no substantial differences in altitudinal distribution between lake-terminating and land-terminating glaciers that could partly explain this difference. The large velocity contrast coincides with a larger surface area of lake-terminating, clean-ice glaciers compared to land-terminating, clean-ice glaciers ($8.4 \pm 6.2$ km$^2$ vs $4.7 \pm 2.0$ km$^2$), and longer ablation areas of lake-

terminating, clean-ice glaciers ($3800 \pm 1601$ m) compared to land-terminating, clean-ice glaciers ($3040 \pm 1127$ m).

**Table 6: Mean of median centre flow line velocities of lake-terminating and land-terminating, subdivided by surface type. Uncertainty estimates represent the 1 SEM confidence interval.**

|  | Terminus Type | | |
| Surface Type | Lake Mean (m yr$^{-1}$) | Land Mean (m yr$^{-1}$) | Both Mean (m yr$^{-1}$) |
| --- | --- | --- | --- |
| Clean-ice | $22.5 \pm 0.52$ | $10.3 \pm 0.14$ | $12.1 \pm 0.18$ |
| Debris-covered | $11.5 \pm 0.63$ | $5.96 \pm 0.11$ | $6.38 \pm 0.12$ |
| All | $18.8 \pm 0.41$ | $8.24 \pm 0.12$ | $9.39 \pm 0.12$ |

Debris-covered, lake-terminating glaciers do not have the same concave-up velocity profile which characterizes their land-terminating counterparts. Their velocity profiles show a larger spread than land-terminating, debris-covered glaciers close to their termini. Lake-terminating, debris-covered glaciers show distinctly enhanced surface lowering very close to their termini, with rates of surface lowering exceeding those on the ablation zone of land-terminating glaciers. Notably, these lake-terminating glaciers are generally smaller in surface area ($6.78 \pm 4.4$ km$^2$ vs $9.4 \pm 6.9$ km$^2$) and the ablation zones are much

shorter ($2720 \pm 1660$ m vs $5680 \pm 3084$ m) than clean-ice, lake-terminating glaciers. Many of the pro-glacial lakes in front of these small, lake-terminating debris-covered glaciers are between 1 and 4 km in length, which likely explains the difference in debris-covered glacier length depending on terminus type.

**4.7 Synthetic Numerical Experiment**

Results from the synthetic numerical experiment indicate that both changes at the frontal boundary condition and in basal

friction can alter the glacier dynamics (Fig. 9), with velocities at the terminus increasing in response to a reduction in effective pressure and in response to lake surface lowering. When the glacier front reaches flotation ($\Delta D = 0$ m), the velocity within ~1 km$^2$ proximity of the terminus increases significantly if the glacier bed friction is dependent on the effective pressure. An acceleration towards the terminus is shown when the ice cliff height increased through a sufficient lowering of the surface lake




level ($\Delta D \geq 10$ m), resulting in a larger frontal imbalance. We find that a smaller cliff height ($\Delta D < 10$ m) only has a limited

effect on the glacier dynamics as the basal drag quickly increases when moving away from a buoyant situation, which is related

to the non-linear sliding law we adopted in this study (m = 3). Beyond a certain increase of the cliff height ($\Delta D > 10$ m), a

further increase of the cliff height above buoyance through lowering of the lake level rapidly increases the surface velocity.

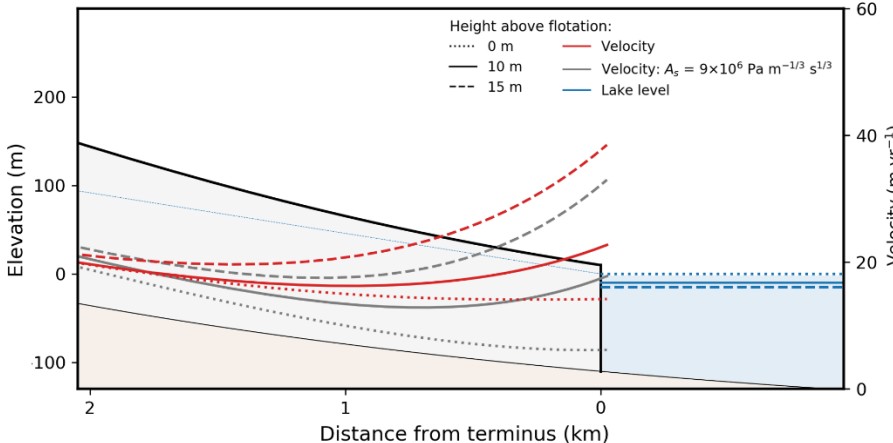

**Figure 9: Velocity results from the numerical experiment for three varying lake levels ($\Delta D$), for both effective pressure dependent**
**roughness parameter and a constant roughness parameter ($A_s$). Blue dotted line indicates the piezometric surface for when $\Delta D = 0$**
**m.**

Our sensitivity experiments show a high dependence on ice thickness (Fig. 10a, b) for both a constant roughness parameter $A_s$

and for a varying $A_s$ to account for the ice thickness uncertainty. This is a direct result of the frontal dynamic boundary

condition, (Eq. 7) which is a function of the ice-thickness. Also, the velocity sensitivity at the terminus to ice thickness might

explain the limited dynamic impact at debris-covered glaciers, which are often relatively thin at their termini. Glacier width or

surface slope also modify the velocity field substantially, but do not heavily influence the relative change in velocity towards

the glacier terminus (Fig. 10c, d).





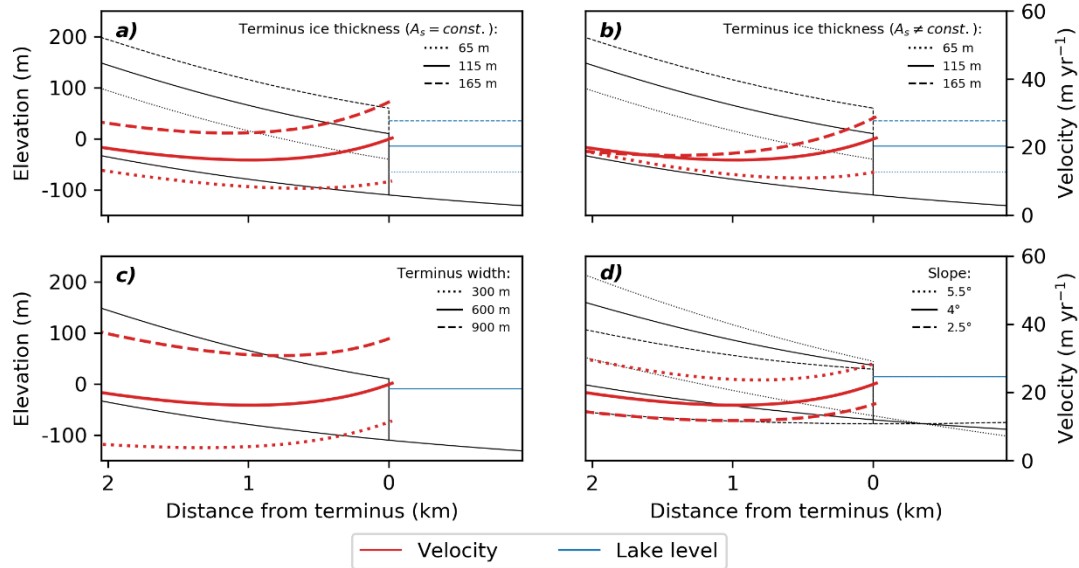

**Figure 10: Velocity sensitivity experiment to ice thickness (a), ice thickness with varying roughness factor (b), terminus width (c) and slope (d).**



## 5 Discussion

### 5.1 Lake-Terminating Heterogeneity in Velocity

The contrasting morphological attributes of lake-terminating glaciers on either side of the main orographic divide likely influences their flow regime and propensity to retreat dynamically. Clean-ice lake-terminating glaciers, mainly found on the north facing slopes of the TP, are larger in size and have a longer ablation zone than clean-ice land-terminating glaciers. Debris-covered lake-terminating glaciers, predominantly found on the southern side of the main orographic divide, are generally much smaller and have shorter ablation zones than their land-terminating counterparts. This contrast in glacier dimensions illustrates how the evolution and occurrence of lake-terminating glaciers must be put in context of the surface cover properties of lake-terminating glaciers, which is related to the morphological settings in which clean-ice and debris-covered glaciers are prone to develop.

Clean-ice land-terminating glaciers with extensive accumulation zones, often flowing onto the TP, possessed enough erosive power to form overdeepenings and large terminal moraines (Scherler et al., 2011). Overdeepenings beneath glaciers flowing onto the TP plateau are not only promoted by terminal moraines but are inherent features of the reversed slope of the TP itself (Royden et al., 2008), making these localities a hot spot of pro-glacial lake development. Therefore, clean-ice lake-terminating glaciers are often large and show consequently higher velocities over the entire ablation length than their land-terminating counterparts.

Unlike for clean-ice glaciers, our results suggest that there is no clear glacier size related preference for proglacial lake development on debris-covered glaciers. Akin to previous studies, we find that low-gradient, debris-covered ablation zones of many Himalayan glaciers (Steiner et al., 2019; Wijngaard et al., 2019) are the result of a concave-up SMB gradient (Bisset et al., 2020), act as a sweet spot for proglacial lake development (Benn et al., 2012; Quincey et al., 2007) and become bounded by a stagnant, ice-cored moraine dam. The development of a proglacial lake leads to a transformation which is associated with a drastic increase in retreat rates (Basnett et al., 2013; King et al., 2019; Watson et al., 2020), and results in lake-terminating glaciers of shorter length than land-terminating glaciers.

Lake-terminating debris-covered glaciers can evolve from the median glacier size land-terminating glacier population, whereas lake-terminating clean-ice glaciers predominantly evolve from land-terminating glaciers that are relatively great in surface area. This, together with the over-representation of clean-ice glaciers in the lake-terminating glacier population (50 out of 70), explains a large part of the lake-land velocity contrast (Fig. 8a). Also, this over-representation of clean-ice glaciers in the lake-terminating glacier population explains a significant part of the contrasting thinning observed in Fig. 8b, which makes it erroneous to attribute this contrast entirely to dynamic thinning. Notwithstanding, the large spread at the terminus of lake-terminating glaciers, with accelerating velocity for almost half of the population (Fig. A4, in the appendix), and elevated surface lowering for both debris-covered and clean-ice lake-terminating glaciers clearly shows that dynamic thinning is a process that must be considered.



## 5.2 Drivers of Dynamic Thinning

The acceleration at the terminus of half of the lake-terminating glaciers in our study area (Fig. A4, in the appendix) strongly indicates an influence of lakes on glacier dynamics. A visual inspection of Fig. 11 underlines the regional extent over which dynamic thinning can be observed, although again, a large heterogeneity in the acceleration at the terminus is visible. Our

numerical experiment indicates that the frontal boundary condition, basal friction, and ice thickness likely play a profound role in accelerating the glacier terminus. However, region-wide measurements in the Himalayas of the ice cliff heights, or ice thickness are absent or highly uncertain, restricting us to a mainly qualitative evaluation of processes that drive dynamic thinning.

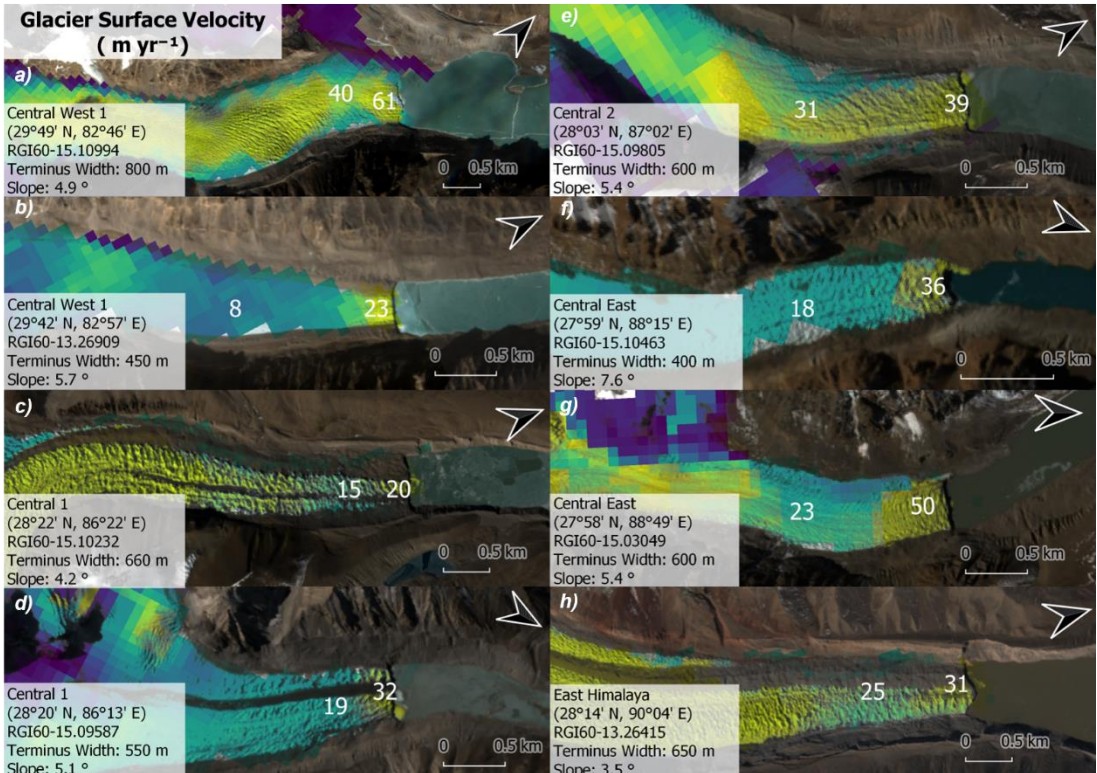

**Figure 11: Examples of lake-terminating glacier accelerating towards their terminus with glacier attributes within 2 km of the terminus. White numbers represent the glacier surface velocity in m yr⁻¹. Colour scale of plotted velocity data is indicative and varies among glaciers. Velocity data and RGB images are retrieved from Sentinel-2.**

A reduction in basal friction caused by the development of a proglacial lake is often designated as one of the main drivers of dynamic thinning (Carrivick et al., 2020; King et al., 2018; Liu et al., 2020; Sutherland et al., 2020; Tsutaki et al., 2013, 2019)

though several remarks must be made when evaluating the importance of this resisting force. Before the development of a proglacial lake, a perched water table is already present due to impermeable moraine at the front. The water pressure at the glacier bed in these overdeepening localities can be expected to depend on this local water table, and the presence of a proglacial lake will therefore be of no influence on the effective pressure if the glacier surface had no time to dynamically





adjust, such as studied by Tsutaki et al. (2019). In that case, the force balance at the glacier front entirely determines the
dynamic influence of a glacier lake and results from a reduction in buttressing caused by a hypothetical instantaneous
expansion of the proglacial lake at the expense of the glacier front.

However, such a situation might be unrealistic, as we have just concluded that a glacier will dynamically respond and thin,
bringing the calving front closer to flotation. The reduction in basal stress then largely depends on the sliding law, for which
we took a non-linear Weertman-type that includes effective pressure dependency (m = 3). Here, basal stress dramatically
decreases when the glacier front approaches flotation which in turn leads to further acceleration of glacier flow (Benn et al.,
2007b). Nevertheless, as our model results suggest, the reduction in the force imbalance at the front through thinning might
dominate in the dynamic response, resulting in an overall decelerating flow when the glacier reaches flotation. This indicates
for a dynamic regime where imbalances at the frontal boundary, through for example glacier retreat rates or a lake-level
lowering, are balanced by enhanced dynamic thinning rates, resulting in a reduced acceleration of the flow, which is
conceptually in line with Nick et al. (2009).

At an early stage of lake development, however, when a clear calving front is yet absent, Tsutaki et al. (2013) showed that a
reduction in basal friction can have a key control on the velocity evolution of the glacier through dynamic thinning and
subsequently promoting the development of transverse crevasses and a disintegration over the expanded area. Also, when the
glacier only thins locally at the calving front, the increase in surface slope might promote an acceleration of the flow again
(Benn et al., 2007a), something this study has not been able to test without violating the SSA's and requires the use of a higher-
order model for further investigation.

The scarcity of local data on glacier ice thickness or lake depth at the calving front makes it difficult to evaluate the
representativeness of the frontal configuration we used in our experimental setup. Watson et al. (2020) found a mean calving
front height of three Nepalese lake-terminating debris-covered glaciers varying between 27 m and 41 m, which indicates that
we can assume with relative high confidence that these glaciers are at least 10 m above flotation, considering the upper-range
ice thickness estimates from Farinotti et al. (2019). Also, for all three glaciers, a surface acceleration in the proximity of the
glacier front was observed (Watson et al., 2020), indicating that the force balance at the boundary condition might dominate
over the importance of an in-situ reduction in basal drag. Remarkably, these lake-terminating glaciers were heavily debris-
covered, indicating that also for these glaciers similar processes are relevant.

Lake-terminating glacier dynamics can only be understood as inseparable parts of an intricately coupled system with frontal
ablation (Benn et al., 2007b), consisting of mechanical calving and subaqueous melt (Carrivick and Tweed, 2013). If frontal
ablation rates are considerably high, dynamic thinning rates are expected also to be high, to bring the ice front closer to a
balanced state. This is in line with annual velocity and glacier retreat observations at the Longbasada Glacier, where Liu et al.
(2020) found the glacier acceleration to be driven by glacier retreat through calving since the mid-1990's. Interestingly, an
above average glacier acceleration was observed in 2006 after 5.6 m surface lake lowering as a mitigation measure in 2005
(Xiao & Dai, 2011). Again, a local increase in the surface slope as a result from dynamic thinning could play a key role here
by promoting calving fluxes under close to buoyant conditions (Benn et al., 2007b).





Another important factor is the reverse gradient of the bed slope, which is an inherent property of Himalayan proglacial lakes (i.e., Somos-Valenzuela et al., 2014), resulting in an instability as the force imbalance at the front will increase with a glacier

retreating to deeper water with greater ice thickness. This phenomenon is known for marine ice sheets as marine ice sheet instability (MISI) (Katz and Worster, 2010; Weertman, 1974), and is shown to be also an important mass loss feedback mechanism in lake-terminating settings (Sutherland et al., 2020). For all these rapid changes, the longitudinal stress gradient plays a large role by distributing the dynamic response triggered at the calving front from the reduction of buttressing (Benn et al., 2007b).

Finally, it needs to be considered that glacier width and surface slope are further relevant factors in the dynamic evolution of lake-terminating glaciers, as local variations of glacier width or high slopes in the close proximity of the terminus are known to be imperative in the context of the transient evolution of these glaciers (Benn et al., 2007b). However, in this study we merely focus on direct processes that accelerate the glacier terminus and drawing conclusions about these factors in a transient context would be outside the scope of this study.

### 590    5.3 Implications for Future Evolution of Himalayan Glaciers

As King et al. (2019) pointed out, regional ice loss through lake-terminating dynamics will remain important in the near future, given the sustained expansion of proglacial lakes across the Himalayan region (Nie et al., 2017; Zhang et al., 2015) and the susceptibility of many debris-covered glaciers for proglacial lake development. Our results also emphasise the importance of clean-ice, lake-terminating glaciers terminating in overdeepening (Linsbauer et al., 2016) and their future contribution to

regional ice loss might be disproportionately large, considering the region wide active flow and their propensity to thin dynamically. Many of these clean-ice glaciers drain northwards into the tributaries of the Brahmaputra, a river of which the melt water supply is of high importance during the dry season (Pritchard, 2019), and changes in ice mass loss projections are of essential importance for millions of people in downstream regions (Immerzeel et al., 2020).

In order to better understand the impact of proglacial lakes onto glacier dynamics and to find out whether the contribution of

lake-terminating glaciers to Himalayan ice mass loss may increase further, more temporal analysis on the glacier-lake dynamics is needed, such as those by Liu et al. (2020) and Watson et al. (2020). In this context, measurements of lake-depth, ice cliff height, ice thickness and surface slope in the proximity of the calving front are essential and will help to constrain factors controlling the dynamics of these lake-terminating glaciers. There is also a need for more detailed modelling studies on glacier-lake dynamics with a particular focus on basal friction during the transient evolution of lake-terminating glaciers in

alpine settings with respect to the importance of the force balance at the glacier front. Finally, considerable progress still can be made by linking frontal mass loss processes that are found to be characteristic for lake-terminating settings (Watson et al., 2020) to glacier dynamics as a fully intercoupled system.





## 6 Conclusions

In this study, we documented 2017-2019 surface velocities in the ablation zone of glaciers larger than 3 km$^2$ by analysing
Sentinel-2 optical satellite imagery at five proglacial lake-prevalent subregions in the Himalayas. Our results show that the
enhanced resolution of Sentinel-2 with respect to Landsat-8 (10 m vs 15 m) enabled the image-matching algorithm to better
resolve the velocity field at small glaciers, and thereby improving the potential for the analysis of glacier centreline velocities.
Analysis of the centreline velocity profiles revealed that lake-terminating glaciers display substantially higher flow velocities
than land-terminating glaciers (18.8 ± 0.41 m yr$^{-1}$ vs 8.2 ± 0.12 m yr$^{-1}$), and that this finding is consistent regardless of the
orientation, glacier size and subregion of the glacier population. The velocity contrast between lake-terminating and land-
terminating clean-ice glaciers is much greater than for debris-covered glaciers, and we showed that a major contribution of the
mean velocity difference can be attributed to the overrepresentation of large clean-ice glaciers in the lake-terminating
population.

Notwithstanding, both clean-ice and debris-covered lake-terminating glaciers show large heterogeneous behaviour at the
glacier terminus, remain dynamically active along the entire flow, and show an accelerating trend for almost half of the glacier
population, revealing that dynamic thinning is prevalent in the Himalayan region. In line with this, we found that a positive
correlation between high terminal velocities and elevated surface lowering is evident for both surface types.

Our synthetic numerical ice-flow model experiment reveals that the surface flow velocity is most sensitive to changes in the
boundary condition at the terminus of a lake-terminating glacier, with variations in basal friction playing a less prominent role
in our model set-up. Rapid changes in terminus position or proglacial lake level could therefore contribute greatly to the
dynamic evolution of the glacier front itself. Further analyses shows that ice thickness plays a large role in the glacier-lake
dynamics, which might contribute to explain the limited dynamic impact at the, often relatively thin, termini of debris-covered
glaciers.

The contribution of ice mass loss from lake-terminating glaciers is unlikely to diminish in the near future, but the exact
contribution to downriver melt water supply in the next decades is still highly uncertain. An improved understanding of lake-
terminating glacier dynamics, both by field observations and numerical studies, is therefore imperative for the future of those
people that depend on a year-round meltwater supply from the major Himalayan rivers.



## Appendices

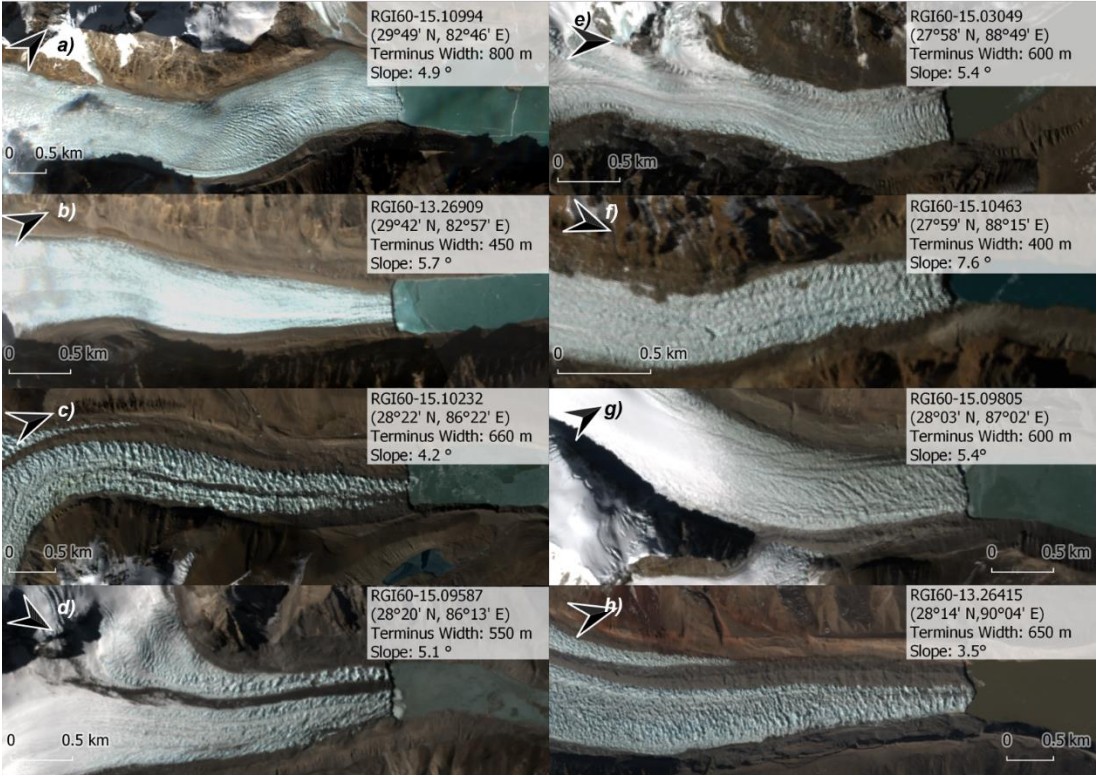


**Figure A1: Examples of lake-terminating glacier with glacier attributes within 2 km of the terminus. RGB images are retrieved from Sentinel-2.**

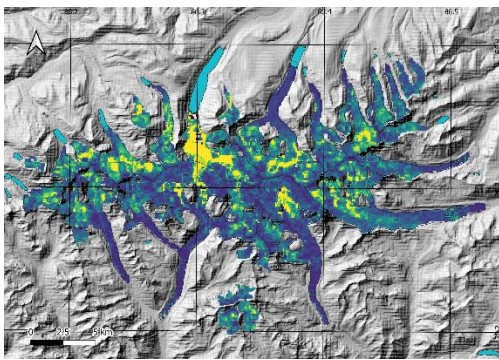

**Figure A2: Velocity dispersion (MAD$_{disp}$). Yellow colours indicate a dispersion >10 m yr$^{-1}$. Hillshade is produced using ALOS World**
**3D DEM.**





**Table A1: Mean of median centre flow line velocities and slope of lake-terminating and land-terminating glaciers for the second half of the ablation zone. Uncertainty estimates represent the 1 SE and 1 SEM confidence interval for the slope and velocity respectively.**

| | Terminus Type | | |
|---|---|---|---|
| Glacier Features | Lake | Land | Both |
| Slope (degrees) | -7.2 ± 3.7 | -8.2 ± 4.54 | -8.0 ± 4.2 |
| Velocity (m yr⁻¹) | 17.7 ± 0.47 | 3.9 ± 0.09 | 5.2 ± 0.11 |


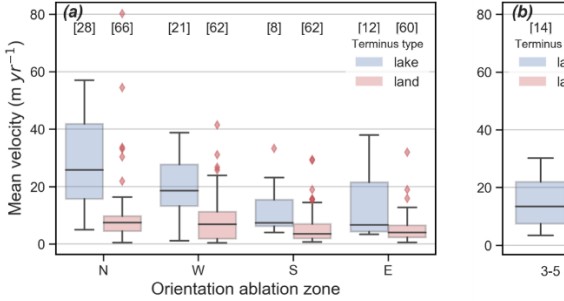
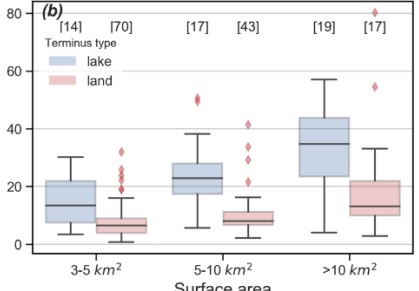

**Figure A3: Boxplot showing the mean velocity contrast between lake-terminating and land-terminating glaciers depending on the orientation of the second half of the ablation zone (a) and surface area (b). Boxes represent the IQR of the distribution. Points outside of the 3rd quartile plus 1.5 times the IQR range are plotted explicitly.**


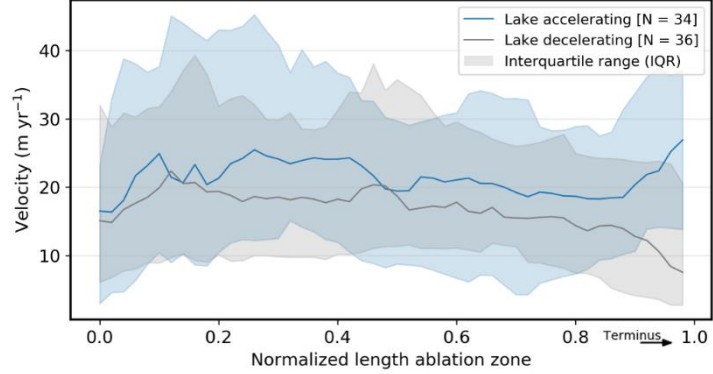

**Figure A4: Separation of accelerating and decelerating lake-terminating glaciers. Glacier is considered to be accelerating if the 500 m in the proximity of the terminus shows higher surface velocities van the 500 – 1500 m proximity range.**






## Data availability

The Central and Eastern Himalaya glacier velocities 2017-2019 (Sentinel 2) dataset will be made available at https://doi.org/10.5281/zenodo.4537289 and www.mountcryo.org.

## Competing interests

The authors declare that they have no conflict of interest.

## Author contributions

J. B. Pronk, T. Bolch and O. King designed the study. J. B. Pronk performed all analysis and wrote the draft of the manuscript. J. B. Pronk, T. Bolch, O. King and D. I. Benn discussed the results and contributed to the writing. All authors edited and contributed to the final form of the manuscript.

## Acknowledgements

We thank F. Maussion for calculating the raw centrelines. We are grateful for T. D. Yao for supporting and G. Q. Zhang for leading field work at Poiqu basin/Central Himalaya. The core of this study is based on a master thesis supervised by B. Wouters and T. Bolch. All processing has been performed using Python and GDAL. This study was supported by the Swiss National Science Foundation (Grant No. IZLCZ2_169979/1), and the Strategic Priority Research Program of Chinese Academy of
Sciences (XDA20100300).



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
