# Peer review of "Proglacial Lakes Elevate Glacier Surface Velocities in the Himalayan Region"

_The Cryosphere, 2021_

## Referee Comment (RC2)

**Review of "Proglacial lakes elevation glacier surface velocities in the Himalayan Region" (tcd-2021-90) by Jan Pronk and others**

**Summary**
The authors undertake an analysis of velocity differences between lake and land terminating Himalayan glaciers. The authors show that lake terminating glaciers are associated with faster ablation-zone speeds than their land terminating counterparts. They then analyze other glacier metrics (e.g, orientation, slope, debris cover) and employ a numerical flow model to aid in the interpretation of their observations. I commend the authors for undertaking such an extensive study that presents some very interesting findings, but find two significant flaws (outlined in "main comments") that must be addressed before the manuscript is ready for publication. I therefore recommend the manuscript undergo major revision.

**Main comments**
1) The language is somewhat stilted in places, with awkward sentence structure and many imprecise/vague statements. The manuscript clarity could benefit from a close read with attention to improving sentence flow and increasing precision and concision. I found much of the writing very dense and difficult to digest and/or follow.

2) You explore covariance of terminus type and several glacier characteristics (orientation, slope, debris cover, etc.). However, it seems that ice thickness differences between the two groups is a very important confounding variable that is not closely considered. As you mention in the text, the lake and land-terminating glaciers have differences in slope, area, and debris cover characteristics, which suggests they would also have difference in ice thickness. If lake terminating glaciers tend to be thicker than land terminating glaciers, this could underlie a substantial fraction of the observed velocity difference between groups. The fact that velocities are close near the ELA suggests that there might not be a gross mismatch, but variations in the distribution of ice thickness between land and lake terminating glaciers could explain the observed velocity differences. This potential complication must be directly addressed. A compelling way to do this would be to utilize the Farinotti 2019 ice thickness product to estimate near terminus ice thickness between these two groups. The fact that calving glaciers do not need ice thickness to go to zero at the terminus is one reason to suspect that lake-terminating glaciers could be thicker, and, hence, faster flowing here. Without investigating this link, you cannot make a casual claim that proglacial lakes cause the observed velocity difference (as is stated in your title), only that the difference exists.

**Minor comments**

L10: the term appears as "proglacial" in the title and "pro-glacial" here. Please be consistent with one use (I think the non-hyphenated version is preferable).

L15: substantially more heterogeneity than what?

L16: effects > affects

L16: it is not clear what you are saying affects half of clean ice glaciers. Dynamic thinning? Terminal velocity heterogeneity?

L41: do you mean "to cause" dynamic thinning?

L54: what do you mean by "rapidly evolving environments"? Vague term that makes the meaning of this clause uncertain.

L55: what is partially decoupled from climate? The transition from land to lake terminating? The

Fig 1: I would suggest using the term "excluded" rather than "uncovered" because of "uncovered" sometimes being used synonymously with "clean" or "debris-free" in a debris covered glacier context.

L100: A reference like Anderson & Anderson, 2016 seems relevant here. Link: https://doi.org/10.5194/tc-10-1105-2016

L138: "The maximum number of image pairs separated by one year was selected for
the month of November, as this month is associated with low cloud cover and a relatively high snow line." – It is unclear what you mean by this. Where is a "maximum number" coming from this analysis? Are you saying you're using November as an end-of-year date?

Sec 3.1.2 & Table 2 -  we need more detail about what velocity fields represent? Rather than just "effective date", it would be useful to know the date of the first and second images used for each correlation. Or at least the midpoint date and the time span between the two images used. Otherwise we don't know if we are seeing annual velocities, seasonal velocities, or some combination. Perhaps this could be visualized as a plot showing the temporal distribution of image pairs for each footprint?

L146-148: does the off-glacier used for estimating coregistraiton error have a similar aspect & slope distribution as the studied glaciers? If not (e.g., steep glaciers & flat area for uncertainty estimate), this error estimate may not be accurate.

L290-291: I am a little confused by this because it seems like basal friction and effective pressure should depend on each other (not be independent as stated).For example, a bed with lower effective pressure will be more slippery (lower friction). Can you better justify this statement or better describe what this experiment is meant to test? If you're not changing sliding rates (through reduced basal traction) by altering lake level, then what exactly are you doing?

L292: exponent in As is not superscripted.

Sec 4.1: How does the absence/presence of debris cover affect velocity uncertainty? Are there systematic differences in debris cover between lake & land terminating glaciers?

Sec 4.2: it seems like the most relevant thing here is whether there are systematic differences between mismatch land & lake terminating glaciers between your estimates & those in ITS_LIVE. If all of your velocities are faster than ITS_LIVE, that doesn't seem like that big of an issue because your study focuses on differences between these groups and is less concerned with absolute accuracy of speeds. However, there would be a problem if lake terminating glaciers are systematically fast biased and land terminating glaciers are slow biased. This analysis should be undertaken.

Table 5: I think you mean ± 4.1 for lake terminating slope (written as 41).

L392: Do you mean "concurrently, IF a large fraction…"? Or are you saying that this is true?

L473: I think you mean 1 km, not 1 km$^2$?

L517: This sentence is fairly awkward and it is hard to determine what you are trying to say.

---

## Author Response (AR1)

**Response by Jan Pronk and others to Anonymous Referee #1 his comment on tc-2021-90.**

*We would like to thank the reviewer for the thorough review and constructive comments. Below we provide our response to each comment. Considerations related to the imagery selection and image processing are especially a reoccurring topic. We would like to start with covering these issues in this introductory response by clarifying the general scope of the study that the authors intended.*

*In this study, our main aim is to show that a contrast in glacier dynamics for land and lake-terminating glaciers is a regional wide phenomenon in the Himalayas. This would be a valuable contribution to the topic of Himalayan lake-terminating glaciers, and lake-terminating glaciers in general, and would be a solid starting point for other studies that investigate the temporal evolution and dynamic drivers of such glaciers, of which the latter is a secondary aim of this study.*

*With this main objective, this study needs to cover all relevant regions where lake-terminating glaciers are prevalent, which, according to Nie et al. (2017), are 5 Himalayan subregions (subdivided by King et al. (2019). To get the best possible results, a large number of image pairs has to be selected. In this study, we only focus on satellite imagery from the month of November. We limit image selection to this month because (1) from April until October the optical satellite imagery is largely obscured by monsoonal cloud cover and (2) from December until April surface contrast is generally low due to low altitudinal, westerly induced, fresh snow cover, making the image matching algorithm perform poorly. Also, after a certain number of image pairs, the reduction of the residual error is only marginal (please see figure 9 from Dehecq et al. (2015)). With an average number of 39 image pairs for each image tile location, we find that our velocity dataset is clearly of adequate quality.*

*Also, the interesting question is raised why we did not investigate the temporal evolution of lake-terminating glaciers by using other available datasets (e.g., ITS_LIVE). This, however, raises several issues that would ask for an entirely different design of this study. Most importantly, when assessing the regional temporal variability, one would likely lose any valuable temporal signal when using a regional wide coverage as individual lake-terminating glaciers will respond dynamically at different times, which would be obscured through the averaging/compilation of results at a regional level. Therefore, a choice has to be made between focussing on the individual glacier and studying the temporal dynamics or focussing on a whole region with a composite dataset, for which we choose for the latter.*

*More in-depth responses to each of the referee his comments can be found below.*

RC: The paper entitled "Proglacial Lakes Elevate Glacier Surface Velocities in the Himalaya Region" aims at quantifying the impact of proglacial lake on glacier dynamics. The paper is divided into two distinct sections. A first part based on remote sensing observations, that compares velocity pattern of lake versus land terminating glaciers, regarding different set of attributes and a second part that models the impact of different parameters on the dynamics of lake terminating glaciers. The main conclusions of the paper are that (1) land terminating glaciers have on average larger ice velocity than land terminating glaciers, (2) high front velocities are correlated with surface lowering, indicating dynamic thinning, (3) ice velocity from lake terminating glacier is mostly sensitive to changes in terminus conditions.

The authors processed satellite data from ESA's Sentinel-2. While their processing chain is based on previous work (Gardner et al., 2020 ; Dehecq et al., 2015) and seems fairly robust, some points are not clear and needs to be more explicit (postprocessing, selection of stable areas, center line analysis, choices of repeat cycles…) and compared with existing studies that uses Sentinel-2 data to map ice flow velocity of glaciers. Specifically, if the aim of the study was to calculate a composite map, I do not understand why the author only choose to process pairs of images separated by 1 year ? Studies have shown, that using all possible repeat cycles and stacking them would largely improve (1) the signal to noise ratio and (2) the spatial and temporal coverage. This would be of great interest for the authors that are looking at glaciers with frontal velocity that rarely exceeds few tens of meters.

*Response:*

*We thank the referee for their careful consideration of the approach we have followed to derive glacier surface velocity data. The reviewer has made a number of valid points regarding data selection which we agree is an important aspect of such a study. We feel we have largely optimised our approach to derive a high-quality velocity field. We outline our prior considerations of the points raised by the reviewer below. In line 138-139 we state:*

*'The maximum number of image pairs separated by one year was selected for the month of November, as this month is associated with low cloud cover and a relatively high snow line'.*

*Indeed, using a large number of repeat cycles improves the temporal coverage. However, after a certain number of image pairs the improvement is only marginal (Dehecq et al. (2015), and at some point, a trade-off must be made between the image matching improvement and the computing time. With an average number of 39 (22-76) pairs for each image tile, we are confident that the residual error of our velocity field is low. (please see figure 9 from Dehecq et al. (2015)). Secondly, the image matching is largely hindered by cloud cover and surface cover conditions. Outside of the month November, the quality and consistency of the imagery decreases drastically because of 1) large cloud cover during the monsoonal months and 2) decreased surface contrast due to snow showers carried on westerlies which can occur across a broad elevation range. Therefore, deploying all other suitably imagery would require a tremendous amount of extra computing time but would not significantly improve the dataset. However, we agree with the reviewer that this information was lacking in the manuscript. Therefore, we will expand the text around line 138-139 to cover the complicating factors that limit the suitable time window for feature tracking.*

> *Solution:*
>
> *Section 3.1.2 has been drastically rewritten according to the respond to the comment above. Please see the revised or the marked-up manuscript.*

I found the measure of the uncertainty on the ice velocity to be somewhat inconsistent throughout the paper: velocity profiles provide a measure of the median and interquartile range, the table are showing the mean and standard error of the mean, and the numbers in the text are not always clear. Something that is even more confusing, most velocity profiles are showing IQR at +/- 10 m, (hence it is difficult to draw conclusion from this), while uncertainty on the velocity in the text rarely exceeds 2 m. Right now, it is hard to tell if the difference in velocity pattern between lake and land glaciers is really over the uncertainty?

*Response:*

*We agree that the way we present the measures is somewhat confusing. Indeed, those measures represent something different:*

- *The Median and the IQR give by no means a quantification of the uncertainty, but instead provide an insight on the 'typical' (median) velocity and spread in the velocity among different glaciers.*
- *The uncertainties presented in the tables are described in section 3.5 provide an uncertainty measure of the along-flowline mean velocities of a group of glaciers, given by the mean and standard error.*

*As a result, these two quantities indicate something fundamentally different: One (median and IQR) tells the reader something about the characteristics of the sample group. The other shows the confidence we have in our along-glacier mean quantities. To relate those two different measures to each other in a better way, we will merge sections 3.2 and 3.5 in the revised manuscript.*

*Solution:*

*We restructured the sections in Chapter 3 from '3.2 Uncertainty of the Velocity Field' onwards to create a more natural flow as follows:*

*Old:*

*3.2 Uncertainty of the Velocity Field*

*3.3 Surface Elevation Change, Estimation of ELA and Surface Slope3.4 Glacier Centre Flow Line Analysis*

*3.5 Glacier Group Uncertainty*

*New:*

*3.2 Uncertainty of the Velocity Field*

*3.3 Glacier Group Uncertainty*

*3.4 Surface Elevation Change, Estimation of ELA and Surface Slope*

*3.5 Glacier Centre Flow Line Analysis*

Concerning the standard error of the mean given in the Table 4, how was this calculated ? The authors need to keep in mind that the ice velocities do not follow a Gaussian distribution, hence using the standard error of the mean does not apply.

*Response:*

*The information is provided (see section 3.5 for the relevant methodology). but we agree that the writing needs to be clarified.*

*We will clarify this issue in the improved version of the manuscript. In short: we sample the background distribution without assuming it is gaussian. From all the resampled means we calculated the 1SE interval, as we assumed this would provide a good indication of the variation of the velocity estimates. However, those sampled means might indeed be (slightly) skewed. We therefore could present this confidence interval by presenting the 1st and 3rd quartile or add a measure of the skewness of the background population to the table with the velocity measurements, as this provides more information about the background distribution.*

> *Solution:*
>
> *As the referee made a valid comment about the uncertainty estimates, we decided to use the IQR as our estimator for the uncertainty spread (see section 3.3). The uncertainty range around the median of mean velocities will therefore be denoted by denoting the first quartile Q1 and the third quartile Q3 behind the velocity estimates. The IQR is also used to quantify the spread among a sample group of a certain variable. The definition of the IQR is explained explicitly for each table and figure.*

All of this is a bit confusing, and things needs to be more homogeneous throughout the text in order to have more confidence in the results.

*Response:*

*We apologise that the information is not presented in a clear way. We will improve the presentation of the methods in the revised manuscript.*

Additionally, the error estimation is largely based on the analysis of velocity fields on stable ground that are selected by the author. However, the selection of these regions is not clear at all throughout the text. Is it selected randomly (including valleys, mountain peaks etc.)? More details need to be provided in this regard, with potential additional figures.

*Response:*

*We agree that more details need to be provided to be able to fully understand the approach. Our approach was the following:*

*For each image tile we selected a square area of which we could reasonably assume the displacement to be zero. We therefore avoided glaciated terrain and high alpine terrain that might be abundant with glacial features such as rock glaciers, as we do not expect these areas to remain fixed through time. We made sure that that the stable area of each image tile was of sufficient width such that it covered multiple granules ($\pm 25km$ in width).*

*We will improve the revised manuscript accordingly.*

> *Solution:*
>
> *We thoroughly have rewritten section 3.1.3, which has been merged with section 3.1.4 and 3.1.5. Here, sufficient attention is given to the procedure related to the stable area selection. Please read the new manuscript for the specific changes made.*

Moreover, despite the fact that the authors processed a large number of Sentinel-2 data, the entire study is based on the analysis of a composite map, hence completely losing the temporal variability in glacier dynamics. I was in turn, a bit surprised to see that the observation part is only based on the comparison of velocity patterns between lake and land terminating glaciers, despite assessing the real influence of lake changes on glacier dynamics.

*Response:*

*As our glacier surface velocity dataset spans a relatively short period (2016-2019), we would suggest that this dataset is not long enough the assess the temporal variability in glacier dynamics. Also, when assessing the temporal variability of the whole region, one would likely lose again any valuable temporal signal when using a regional-wide coverage as individual glaciers will respond dynamically at different times, which would be obscured when collated or averaged out at the regional level. Therefore, a choice has to be made between focussing on the individual glacier and studying the temporal dynamics or focussing on a whole region with a composite dataset. This complicates the possibility to address the temporal variability, as our focus of the study is regional. Moreover, Dehecq et al. (2019) discuss in the section 'Ice dynamics response to thickness change' a lag of a few years between the thickness change of a glacier and driving stress. This lag would impact the robustness of velocity change analyses over the time period covered by our velocity field.*

While the dataset from ITS_LIVE is less resolved than this dataset, it would have been exciting to monitor changes in ice dynamics directly related to changes in lake heights (measured from altimetry for example) or lake area (from optical or SAR imagery), during the last 20 years. As a consequence, this would have been directly related to the second part of the paper that is modeling the influence of several parameters on glacier dynamics.

*Response:*

*We appreciate the suggestion of the referee and agree that this would be interesting information. An investigation of lake heights and areas for a large sample over a multi-temporal time frame would be something worthwhile for future studies. For now, we think that it is just as important to characterise the differences in dynamics relating to terminus type on a regional wide scale such as done in this manuscript, which has not been done before and paves the way for a multi-epoch approach such as the referee suggests. As such, we think that this suggestion is beyond the scope of the study, but needs to be addressed in the discussion, which will be done accordingly. Moreover, we would like to point out that within such a study set up, one would lose most valuable insights of the temporal variability when assessing lake-terminating glacier surface velocities on a regional wide scale. One would have to focus on specific glaciers again (Tsutaki et al., 2019), losing the regional scope of this paper.*

This brings me to my final point: I found the two main sections of the paper a bit disconnected from each other's. The first one investigates the relation between several parameters (glacier orientation, area, debris cover) and velocity patterns, but there is little in common with the second section that really deal with what processes that are influencing lake-terminating glacier dynamics (where there is also no comparison with land terminating glaciers). This raises a number of interesting questions.

*Response:*

*We agree that the two parts could have been better connected. We are convinced that the modelling section provides useful additional insights where the drivers of the observational section are investigated. A comparison to a land-terminating glacier has already been conducted by several other studies (e.g., Tsutaki et al., 2019) and would not necessarily add new insights to this field. Nevertheless, we will work on improving the revised manuscript to better connect the main sections together.*

   *Solution:*

   *We thoroughly worked on improving the connection between the remote sensing and the modelling part. We mainly improved the flow by rewriting the start of section 3.6.1 and 4.7 to put the modelling part in perspective of the broader objective of this study. Please see these specific sections in the new manuscript.*

Indeed, is it right now possible to observe the influence in proglacial lake changes on changes glacier dynamics (ex: changes in lake level to be consistent with the modeling section)?

*Response:*

*We agree with the author that this is an interesting line of investigation. Our study focusses on the diagnostic differences between land- and lake-terminating glaciers, and we therefore have no observational data on temporal changes of proglacial lakes. Although there is only limited observational evidence, in section 5.2 we mentioned observed changes of the frontal boundary condition followed up by changes in glacier dynamics:*

*In line 576-578 we wrote:*

*"Interestingly, an above average glacier acceleration was observed in 2006 after 5.6 m surface lake lowering as a mitigation measure in 2005 (Xiao & Dai, 2011)."*

Is there data available to see the formation of proglacial lakes, and how a change in the area of those lakes have influenced surface flow velocity?

*Response:*

*There is certainly a robust relationship between lake area and lake depth, and consequently the area of contact between the lake and glacier terminus, which might alter the boundary conditions. However, our results indicate that the relation between lake area and surface flow velocity is ambiguous and likely lacks a direct causal relation. This should be investigated much more thoroughly, but the paucity of freely available lake bathymetry measurements across the Himalaya make this difficult to examine in detail.*

Can we replicate these observations with the models presented in the last section?

*Response:*

*Unfortunately, the accurate replication of specific examples of glacier behaviour would require a comprehensive dataset of lake bathymetry and temporal changes of the lake depth in the direct proximity of the glacier. With most of this data being yet unavailable we feel that this goal would be more suited for studies in the future.*

With the large quantity of optical data to map lakes and already available velocity fields (ITS_LIVE, Golive, Dehecq et al., 2019), I think that it might be possible to assess.

*Response:*

*We thank the referee for his suggestion. Please also see our introductory response. We feel that the quality of those datasets is not good enough to assess temporal variability on a regional scale at the terminus of glaciers $< 5$ km$^2$. One then would have to focus on the larger glaciers, which would result in losing one of the main points of this paper, namely that special terminal glacier-lake dynamics are a regional wide phenomenon. With the improvement of satellite imagery, especially since 2016, a regional wide approach will become slowly but steadily possible.*

*Also, we would again like to point out that within such a study set up, one would lose most valuable insights of the temporal variability when assessing lake-terminating glacier surface velocities on a regional wide scale, as individual glaciers will respond dynamically at different times.*

Finally, it is not clear to me why the author restricted their study area to the central Himalayas. It excludes an entire section of the Himalayas that is very dynamic, and with a lot more diversity in terms of glacier size, orientation, slope and velocity magnitude.

*Response:*

*We appreciate the authors comment and will clarify these considerations properly in the revised manuscript. Please also see our introductory response. We restricted our analyses to these five regions in the central and eastern Himalayas for two reasons. Firstly, outside of the Himalayan regions of this study, the number of lake-terminating glaciers is currently limited (Nie et al., 2017). Consequently, these areas fall outside of our area of interest, as they hardly contribute to an increase in the sample size of lake-terminating glaciers. Secondly, we restricted our data to the extent of the dh/dt dataset of King et al. (2019) to be able to directly compare our velocity dataset to elevation change.*

Figures and Tables are overall clear and well presented.

*Response:*

*We thank the referee for this comment.*

Please find specific comments below:

L77. What do you mean by lake-driven changes in the velocity field ? is it related to the modeling part. Please present the objectives of this paper in the same order as it appears within the text.

*Response:*

*We thank the referee to point this out and agree that this sentence might be confusing for the reader. In the objectives paragraph (around line 77), we write:*

*"More specifically, we seek to investigate the attribution of lake-driven changes in the velocity field to dynamic thinning and investigate the role that debris cover plays on glacier-lake dynamics."*

*In response to this, we will change this to simply 'changes in the velocity field' as one of the main aims is to investigate the prevalence of dynamic thinning, which is inherently lake-driven. We hope that with this, it is clear that the objective is entirely devoted to the remote-sensing part.*

> *Solution:*
>
> *In 76 – 77 we wrote:*
>
> *"More specifically, we seek to investigate the attribution of lake-driven changes in the velocity field to dynamic thinning and investigate the role that debris cover plays on glacier-lake dynamics."*
>
> *We changed this to:*
>
> *"More specifically, we seek to investigate the attribution of changes in the velocity field to dynamic thinning and investigate the role that debris cover plays on glacier-lake dynamics."*

L102. Does that really make a difference ? Mean area of lake terminating glacier is 7 km2. How much glaciers are you adding up with these? Be more quantitative.

*Response:*

*We thank the referee to point this out. In this study we only focus on glaciers with an area larger than 3 km2, compared to the 5 km2 previously utilised by Dehecq et al. (2019). Figure 4 shows that the quality of the dataset clearly improves with the use of 10m sentinel-2 imagery which allowed us to incorporate smaller glaciers into the dataset. Within the 3 to 5 km2 glacier area size group, we identified 23 lake-terminating glaciers, whereas the total lake-terminating dataset constitutes of 70 lake-terminating glaciers.*

> *Solution:*
>
> *In line 106-107 we write: "In this study we only focus on glaciers with an area larger than 3 km2, compared to the 5 km2 previously utilised by Dehecq et al. (2019), which enables us to add substantially more glaciers to our dataset (Fig. 6b).". Here, we refer to Fig. 6b, where the added values of including smaller glaciers is clearly illustrated.*

L105. What do you mean by "very low surface velocity" compare to what and where on the glacier ?

*Response:*

*We agree with the referee that this has not been made clear in the manuscript. Generally, glaciers with a maximum velocity below 10 m/yr. We will make this clear in the text.*

*Solution: We removed the relevant words at line 110 as it did not add any essential information to this section.*

L108. Why do you restrict your study to the Central Himalayas ? By doing so, you are excluding all the glaciers in the Pamir-Karakoram, that are really diverse in terms of size, velocity magnitude, slope, debris coverage… Here you restrict yourself to glacier to mostly small size and slow-moving glaciers, which limits general conclusions that can be made.

*Response:*

*Please see also our introductory response. We restricted our study area to the five regions in the central and eastern Himalayas for two reasons. Firstly, outside of the Himalayan regions of this study, the number of lake-terminating glaciers is limited. Consequently, these areas fall outside of our area of interest (the link between proglacial lakes and elevated glacier flow), as they do not contribute to an increase in the sample size of lake-terminating glaciers. We also assembled our velocity dataset to match the extent of directly comparable dh/dt data of King et al. (2019) to be able to directly compare our velocity dataset to elevation change.*

L 129. How does it compare to the geolocation error calculated by Millan et al., 2019 ?

*Response:*

*The two errors are essentially the same but are simply based on a different approach. Where Millan et al. (2019) use an average (for which we assume to be the median), we use the $95^{th}$ percentile reported by the quality report of Sentinel-2. Millan et al. (2019) finds a average error of 0.52 pixels, which roughly translates into our $95^{th}$ percentile error of 12m. We tend to follow the way of reporting the quality of Sentinel-2 by the official quality report of Sentinel-2 itself, as done in line 129.*L132. Do you mean removing the average offset calculated off glaciers? Which is mentioned in the post-processing?

*Response:*

*Yes, this is a relatively common procedure with Sentinel-2 imagery.*

L 138. Why do you restrict yourself to image pairs at 1-year interval ? Using all pairs of at least >1 month, would greatly improve your signal to noise ratio, which is really important when looking at small velocity numbers (<30 m/yr.) (cf Millan et al., 2019)

*Response:*

*Please see our general response.*

>*Solution:*

>*We cover this now in depth in section 3.1.2.*

Table 2 and L. 141. I don't understand this effective date. Do you mean the central date between image pairs ? Why is it always 2018 ? I thought you processed all data between 2016 and 2019.

*Response:*

*We processed all data between 2016 – 2020 from November that was of adequate quality. If for example, data from 2016 cannot be used, the velocity data will shift to a central date more towards 2019. We thank the referee for the suggestion to use, 'central date' instead, and will do so in the revised version.*

> *Solution:*
>
> *On all relevant localities (table 2, line 148, 149, 236, 237 and 355) we changed 'effective date' for 'central date'.*

L 147-148. How is the stable ground area selected ? Is it random ? Hence including both mountain peaks (potential higher orthorectification errors) and valleys ? 300 km2 is a bit limited to see potential deviation across images and to study noise, specifically with a low number of image pairs. Furthermore, you discuss this also in section 3.1.5 right ? Please remove this part and discuss it later in the appropriate postprocessing section.

*Response:*

*Please do also see the previous general response on the stable area selection. The selection of a stable area is not entirely random. One must be confident that most velocity estimates are zero. Very steep mountains do indeed cause potential higher orthorectification errors, and therefore such areas are generally omitted. The slopes of the glaciers do fall far out of this category, and we therefore argue that those potential higher orthorectification errors are not representative. We thank the referee for his suggestion, and we will remove this part in line 147 – 148 to be clearer with our description.*

> *Solution: We removed the relevant lines at 'section 3.1.3' (this section is merged with 3.1.4 & 3.1.5, see the following comments, responses & changes).*

L 151. Do you calculate gradient in the x and y direction ? Please specify.

*Response:*

*We agree that this could have been specified in the text. We indeed calculate the gradient in the x and y direction.*

> *In line 151 we wrote:*
>
> *"Each pixel represents the orientation of intensity gradient at that pixel, making the method invariant to illumination change, which is a desired property for feature tracking algorithms."*
>
> *We changed this to:*

*"Each pixel represents the orientation of intensity gradient in the x and y direction at that pixel, making the method invariant to illumination change, which is a desired property for feature tracking algorithms."*

Section 3.1.4. This section is too technical and do not bring anything substantially new. The cross-correlation technique has already been widely documented in the literature. Hence, I would suggest to reduce this section and remove equations.

*Response:*

*We thank the referee for his comment. We will condense the three processing sections and refer to existing literature in case of established methods.*

> *Solution: We condensed section 3.1.3, 3.1.4 and 3.1.5, and removed parts that were too technical.*

Section 3.1.5. See previous comment.

*Response:*

*See previous response.*

Section 3.2. See previous comment on the error estimation. Please be consistent throughout the text between IQR, SEM, MAD….

*Response:*

*We agree that the text has to be clearer about these intervals. Therefore, we will improve this in the revised version.*

> *Solution:*
>
> *Please see also our solution provided as a respond to the general comments. In general, we worked thoroughly to make the definition of the error estimators more robust. We now moved towards using IQR for our most robust primary estimator for the spread and uncertainties of relevant quantities. Throughout the manuscript, the definition of this quantity is consistently explicitly given.*

L 216. Is the use of such a large filter size limited compared to the width of the glaciers ?

*Response:*

*The resolution of the velocity field is 80m and filtering therefore incorporates a larger surrounding area. We agree that for the very small glaciers, a filter window of 240 is still large. For this reason, indeed there is more weight on the velocity points on the very centre (see equation 4).*

L 218. By how much does it increases the overall confidence ? Be more specific.

*Response:*

*We weighted estimates with high confidence more: for example, if a neighbouring estimate next to (80m) the centreline has a much higher confidence, weighting this estimate more increases the confidence in the overall velocity estimate. From another perspective: If a centreline velocity estimate has a very low confidence, if might be worthwhile to look at the neighbouring estimate. Whether this increases the confidence drastically depends on each specific site, but it ensures that we retrieve data with the highest confidence possible.*

> *Solution:*
>
> *We checked this: the mean confidence for all glaciers over the ablation zone improved by 22%, whereas the confidence for lake-terminating glacier and land-terminating glacier improved by 24% and 21% respectively.*
>
> *We added to line 339 – 341: "The approach of applying a gaussian window to the velocity estimates reduced the mean CI95 of lake-terminating and land-terminating glaciers by 24% and 21% respectively."*

L 256-261. How did you calculate the A value ? Does it vary spatially ? or do you take one value for the entire glacier/region ?

*Response:*

*We feel that this is clearly written out in the text, with adequate referencing for more in depth information.*

*We wrote in line 256-261:*

*"A is the temperature-dependent rate factor and increases from a minimum of $3.5 \times 10^{-25}$ $Pa^{-3}\ s^{-1}$ at the divide to a maximum of $1.7 \times 10^{-24}\ Pa^{-3}\ s^{-1}$ at the calving front, corresponding to a depth-averaged ice temperature range of $-10°\ C$ to $-2°\ C$ at the ablation zone (Cuffey and Paterson, 2010), for which we follow Enderlin et al. (2013)."*

*However, we can add more detailed information if the reviewer still thinks it would be beneficial.*

L 275-277. Where are these thickness values coming from ? What do you mean by in line with Farinotti et al., 2019 ? You didn't use the value of the thickness for these specific glaciers provided by Farinotti et al ?

*Response:*

*We looked at the thickness of the larger, clean-ice, lake-terminating glaciers in the dataset of Farinotti et al. (2019) and used this as a rough indication of the ice-thickness in our modelling study.*

L 278. How was the piezometric surface assessed ? Provide method and reference.

*Response:*

*The piezometric surface is chosen in such a way that is starts at the lake's water table and slowly slopes upwards up-glaciers, so that it achieves a good fit with observed velocities, for which we follow Benn et al., 2007 (as referred to in the text).*

L 280. What do you mean by up-glacier velocity ? Is it the ablation or accumulation zone ? Be more quantitative. Where is the speed value of 50 m/yr coming from ? Has it been taken from the measured velocity data ? Specify.

*Response:*

*We thank the referee and agree this has to be more explicitly specified. Within our synthetic model set-up, we used 50 m/yr as a rough indication for a maximum velocity of a larger clean-ice lake-terminating glacier typically found at the end of the accumulation zone, which we based on our own velocity dataset. We will specify this in the revised version.*

> *In line 280 we wrote:*
>
> *"We then tuned the sliding parameter ($A_s$) such that the maximum up glacier velocity reaches a typical value of 50 m yr$^{-1}$ and found a value of $A_s = 2.5 \times 10^6$ Pa m$^{-2/3}$ s$^{1/3}$."*
>
> *We changed this to:*
>
> *We then tuned the sliding parameter ($A_s$) such that the maximum velocity near the ELA of the larger clean-ice lake-terminating glaciers reaches a typical value of 50 m yr$^{-1}$ (Dehecq et al., 2019a; Gardner et al., 2020; Pronk et al., 2021), and found a value of $A_s = 2.5 \times 10^6$ Pa m$^{-2/3}$ s$^{1/3}$.*

L 285. Ice thickness from the consensus estimate?

*Response:*

*The referee is correct. We make this more clear by referring to $H_t$ in section 3.6.1 already, where we will write (line 275-277):*

> *Solution:*
>
> *In line 291 (revised Manuscript) we write:*
>
> *"We used a maximum ice thickness (H) of 230 m and an ice thickness of 120 m at the terminus ($H_t$), values in line with ice-thickness estimates of the larger Himalayan glaciers (Farinotti et al., 2019)."*

L 288-289. Be more specific. What is the realistic range ?

*Response:*

*We thank the referee for his comment. However, we think that with a clear reference of Watson et al., (2020), no range needs to be explicitly mentioned, as this only might cause confusion for the reader. In the discussion in section 5.2 we do mention these values when they are useful to be mentioned in the context of that discussion.*L 300. How do you change the ice thickness estimate ? Uniformly ? By how much ? How come you keep the maximum velocity at 50 m/yr. ? Do you still conserve mass ?

*Response:*

*We will address these questions in the last paragraph of section 3.6.2 to make things clear. We do change the ice thickness uniformly by 50 m, as mentioned in the text. To keep the maximum velocity at 50 m/yr, we tuned the sliding parameter $A_s$. We will mention this in the text as well.*

> *Solution:*

> *In line 356 we added 'uniformly' for clarification.*

L 305. Do you mean accuracy or precision ? I think you mean precision here.

*Response:*

*The referee is right, but we realise that precision is also a bit misplaced here.*

*We will rename section 4.1 to 'Algorithm Performance'.*

> *Solution: We renamed section 4.1 to 'Algorithm Performance'.*

Section 4.2. Please provide a figure illustrating the differences between each velocity dataset.

*Response:*

*We thank the referee for his suggestion. We would like to mention that we provide a clear illustration of this in Figure 4.*

Section 4.4. Considering the very large IQR, I do not find any significant differences between lake and land terminating glaciers that is above the noise. Please include Fig A3 in the main text. I think it provide more concluding evidence than Figure 6.

*Response:*

*In the context of earlier comments about the confusion of uncertainty and the IQR, we see how these could be confused . As we now plan to merge the respective sections of the text which describe the IQR and uncertainty (3.2 and 3.5) we feel that the reader will be able to more clearly distinguish between the two in the amended manuscript.*

> *Solution:*

> *The relevant sections are not merged, though put right after each other to improve the flow of the text.*

Figure 4. can you provide error bars for the velocity profiles ? In order to be consistent with the other figures.

*Response:*

*We thank the referee for his suggestion. We think that it is clear from our previous responses that the IQR's shown in the figures are not error bars but show the variability of the background glacier population. We feel that it is important to show the spread of velocity values amongst glacier groups of different terminus type in Figure 4 and we would rather refrain from changing this figure. However, we will prepare this figure with IQR's and add it to the revised manuscript if we find that it does not obscure the original measures.*

Section 4.6. Now this is a bit confusing, some much time have been dedicated to the comparison between lake and land terminating glaciers, but here we left off the land terminating ones. Why is that? Another aspect that would add even more value to the study would be to check out the influence of debris thickness on glacier velocity pattern and magnitude (check out Rounce et al., 2021 for the dataset).

*Response:*

*This section is written in the context of both lake- and land-terminating glaciers, which we would argue is clearly illustrated in figure 8 (blue colours for lake-terminating glaciers throughout all the figures and red for land-terminating). To avoid any confusion, we will edit this section and make sure the differences are clear. We thank the referee for his suggestion on debris thickness, though we feel this would be outside the scope of this paper. Also, it is well documented that the majority of debris covered glacier area across the central Himalaya are stagnant or flowing below the level of detection of feature tracking algorithms. We do not feel that this needs re-emphasising.*

> *We think that within the context of the improved manuscript, the context of this section will be clearer.*

L 504-509. The relation between the large-scale evolution of the Tibetan plateau and the formation of over deepening is not clear to me at this point. Please be more specific.

*Response:*

*To make things clear, we will mention in the revised version that towards the Tibetan Plateau, the elevation generally slopes upwards (promoting overdeeping) (Royden et al., 2008).*

> *We clarified this in the new Manuscript by writing in the introduction section in line 64-66: "The number and total area of proglacial lakes in the Himalayan region has increased (Nie et al., 2017; Shugar et al., 2020; Zhang et al., 2015), a trend which is likely to continue in the near future, as many glacier beds are characterised by overdeepenings (Linsbauer et al., 2016). "*

L 510-514. Please provide a reference to a figure and section of your paper.

*Response:*

*We thank the referee for his comment and will adopt this in the revised manuscript.*

Figure 11. Please provide a colorbar. Displaying the velocity on a log scale would enable to better observe the acceleration at the ice front which is not always clear (a, c, d). Would also be good to show for each glacier maps of surface lowering (Brun et al., 2017 for example).

*Response:*

*We will provide colorbars in the revised version. If a log scale would indeed improve the referee's suggestion, we will do this as well. Directly comparing the glacier velocity might be interesting but can be problematic as you need a dataset with the same temporal coverage to get matching results for individual glaciers. For this reason, we think that this would be outside the scope of this paper.*

> *Solution:*
>
> *We tried displaying the velocity on a log scale, which did not improve the quality of the figure. Also, we found that the white numbers in Fig. 11 remained the best solution to clarify the absolute magnitude of the velocity profiles.*

L 517-520. It is not clear to me where this is going. Split this sentence into one or two difference sentences to make your argument clearer.

*Response:*

*We thank the referee for his comment and will follow his suggestion to create more structure.*

> *We changed (line 517 – 520): Lake-terminating debris-covered glaciers can evolve from the median glacier size land-terminating glacier population, whereas lake-terminating clean-ice glaciers predominantly evolve from land-terminating glaciers that are relatively great in surface area. This, together with the over-representation of clean-ice glaciers in the lake-terminating glacier population (50 out of 70), explains a large part of the lake-land velocity contrast (Fig. 8a).*
>
> *Into (line 539-542):*
>
> *"As a result, lake-terminating debris-covered glaciers develop from the glaciers whose area is close to the median of the land-terminating glacier population, whereas lake-terminating clean-ice glaciers predominantly evolve from larger land-terminating glaciers. This, together with the over-representation of clean-ice glaciers in the total lake-terminating glacier population (50 out of 70), explains a large part of the lake-land velocity contrast (Fig. 8a)."*

L 530. What about the lake temperature ? Could we potentially imagine that rising up the lake temperature would increase the melt at the front of the glacier and triggers an acceleration, as it is observed in Greenland and Antarctica ?

*Response:*

*In the revised version, we will mention the potential importance of the lake-temperature as potential ultimate driver of changes in the boundary condition.*

*We felt that this would be slightly outside of the scope of this paper and decided to not include this in the revised manuscript.*

L 609. Change 2017 to 2016 ?

*Response:*

*First images are from November 2016. Matching these with images from November 2017 results in a velocity field at mid-2017.*

*References:*

*Dehecq, A., Gourmelen, N. and Trouve, E.: Deriving large-scale glacier velocities from a complete satellite archive: Application to the Pamir-Karakoram-Himalaya, Remote Sens. Environ., 162, 55–66, https://doi.org/10.1016/j.rse.2015.01.031, 2015.*

*Dehecq, A., Gourmelen, N., Gardner, A. S., Brun, F., Goldberg, D., Nienow, P. W., Berthier, E., Vincent, C., Wagnon, P. and Trouvé, E.: Twenty-first century glacier slowdown driven by mass loss in High Mountain Asia, Nat. Geosci., 12(1), 22–27, https://doi.org/10.1038/s41561-018-0271-9, 2019.*

*Nie, Y., Sheng, Y., Liu, Q., Liu, L., Liu, S., Zhang, Y. and Song, C.: A regional-scale assessment of Himalayan glacial lake changes using satellite observations from 1990 to 2015, Remote Sens. Environ., 189, 1–13, https://doi.org/10.1016/j.rse.2016.11.008, 2017.*

*Royden, L. H., Burchfiel, B. C. and Van Der Hilst, R. D.: The geological evolution of the Tibetan plateau, Science (80-. )., 321(5892), 1054–1058, https://doi.org/10.1126/science.1155371, 2008.*

*Tsutaki, S., Fujita, K., Nuimura, T., Sakai, A., Sugiyama, S., Komori, J. and Tshering, P.: Contrasting thinning patterns between lake- And land-terminating glaciers in the Bhutanese Himalaya, Cryosphere, 13(10), 2733–2750, https://doi.org/10.5194/tc-13-2733-2019, 2019.*

**Response by Jan Pronk and others to Anonymous Referee #2 his comment on tc-2021-90.**

**Summary** The authors undertake an analysis of velocity differences between lake and land terminating Himalayan glaciers. The authors show that lake terminating glaciers are associated with faster ablation-zone speeds than their land terminating counterparts. They then analyze other glacier metrics (e.g, orientation, slope, debris cover) and employ a numerical flow model to aid in the interpretation of their observations. I commend the authors for undertaking such an extensive study that presents some very interesting findings, but find two significant flaws (outlined in "main comments") that must be addressed before the manuscript is ready for publication. I therefore recommend the manuscript undergo major revision.

*We thank the reviewer for his/her thorough assessment of our study and for the positive comments on our results so far. We agree with the reviewer's points about the benefits of an assessment of the relationship between glacier surface velocities and ice thickness over the glaciers in our sample. Indeed, we had conducted a prior analysis of the relationship between our surface velocity results and the ice thickness estimates of Farinotti et al. (2019). Although this ice thickness data has shown to be of great value for regional ice volume estimates, we initially decided against including these results in the paper because of the large uncertainties inherent with the ice thickness data when considering the ice thickness distributions as a stand-alone variable. Therefore, we feel that for the purpose of ice thickness evaluation along a flowline, such ice thickness data should be interpret with caution. For the revised manuscript, we will put nevertheless more emphasis on this important variable and also consider showing the thickness figure in the supplement.*

**Main comments**

1) The language is somewhat stilted in places, with awkward sentence structure and many imprecise/vague statements. The manuscript clarity could benefit from a close read with attention to improving sentence flow and increasing precision and concision. I found much of the writing very dense and difficult to digest and/or follow.

*Response:*

*We thank the referee for his/her comment. We will go through the manuscript to improve the readability, focussing in particular on the clarity of the writing.*

> *Solution: all the authors contributed by improving the readability of the text.*

2) You explore covariance of terminus type and several glacier characteristics (orientation, slope, debris cover, etc.). However, it seems that ice thickness differences between the two groups is a very important confounding variable that is not closely considered. As you mention in the text, the lake and land-terminating glaciers have differences in slope, area, and debris cover characteristics, which suggests they would also have difference in ice thickness. If lake terminating glaciers tend to be thicker than land terminating glaciers, this could underlie a substantial fraction of the observed velocity difference between groups. The fact that velocities are close near the ELA suggests that there might not be a gross mismatch, but variations in the distribution of ice thickness between land and lake terminating glaciers could explain the observed velocity differences. This potential complication must be directly

addressed. A compelling way to do this would be to utilize the Farinotti 2019 ice thickness product to estimate near terminus ice thickness between these two groups. The fact that calving glaciers do not need ice thickness to go to zero at the terminus is one reason to suspect that lake-terminating glaciers could be thicker, and, hence, faster flowing here. Without investigating this link, you cannot make a casual claim that proglacial lakes cause the observed velocity difference (as is stated in your title), only that the difference exists.

*Response:*

*We agree with the referee that ice thickness data are an essential variable that ideally must be considered. We initially did analyse the Farinotti et al. (2019) ice thickness product in detail to explore its potential. A limitation of this data however is that it comes with large uncertainties. This might be especially true at the glacier termini and glaciers with debris cover, where errors might be systematic due to the methodology by which ice thickness data is calculated, which heavily depend on SMB assumptions. Also, no information is available on whether uncertainties are systematic near the terminus of proglacial lakes. However, we fully agree that attention must be given to the general importance of this parameter.*

*Figure 1 (see this supplement) shows the median ice thickness of all land- and lake-terminating glaciers in our sample group, based on the ice thickness dataset of Farinotti et al. (2019). It shows that, from the middle part of the ablation zone onwards, the velocity contrast between lake- and land-terminating glaciers might be (partly) attributed to differences in ice thickness. Nevertheless, the data also shows a clear decrease in ice thickness for both land- and lake-terminating glaciers towards the termini. At the same time, the lake-terminating glacier velocity does not show a decrease towards the terminus, and even accelerates for half for the glacier sample group (see Figure 5). This indicates that ice thickness data is important and must be considered but cannot explain the whole velocity contrast at the glacier terminus.*

*Also, the authors are correct in stating that lake-terminating glaciers are thicker near the terminus than terrestrial ones, due to the fact that they end at a calving cliff rather than a front that thins to zero. This may indeed influence the velocity. However, this difference in ice-thickness is also due to the presence of a lake. Although there is with this mechanism no direct positive feedback link by which ice mass loss is enhanced, it does describe a clear causal relation between the presence of a lake and elevated terminal velocities. We will formalise this when discussing mechanisms by lakes encourage higher velocities through the: 1) force imbalance at the terminus; 2) elevated water pressures; and 3) non-zero ice thickness at the terminus.*

*For the revised manuscript, we will consider showing the thickness figure in the supplement and devote some text to this important variable.*

    *Solution:*

    *We agreed with the referee that ice thickness must be considered. We therefore added an entire paragraphs (line 594 – 609) to section 5.2 in the revised manuscript:*

    *"The frontal ice thickness itself is a variable that needs more consideration when evaluating drivers of frontal ice velocity. Evidently, lake-terminating glaciers are thicker near the terminus that land-terminating glaciers, since they end at a calving cliff rather than at a front that thins to zero. As ice thickness drives ice flow (see the*

*right-hand side of Eq. (5)), a substantial part of the terminal velocity contrast between lake-terminating and land-terminating glaciers could then be attributed to this difference in ice thickness. Indeed, comparison of the median ice thickness of our glacier sample group (Fig. A5), using the ice thickness estimates from Farinotti et al. (2019), indicates that lake-terminating glaciers are substantially thicker near the terminus. As such, this indicates that ice thickness is a significant factor in determining the frontal ice-flow velocity. However, Fig. A5 also shows a clear decrease in ice thickness for both land-terminating and lake-terminating glaciers towards the terminus. At the same time, the lake-terminating glacier velocity does not show a deceleration towards the terminus, and even accelerates for half for the glacier sample group (Fig. 5). This indicates that ice thickness data is unable to explain the whole velocity contrast at the glacier terminus. Concurrently, it is worth considering that the difference in ice thickness between land-terminating and lake-terminating glaciers is also due to the very presence of a lake. Whilst this suggests no direct positive feedback mechanism is displayed by which ice mass loss is enhanced through dynamic thinning, a causal relation between the presence of a lake and elevated terminal velocities can still be inferred. Errors in these ice thickness estimates are significant and could be systematic depending on surface type, which might be especially true near the terminus. Therefore, these results should be treated with caution until direct measurements of terminus ice thickness are available."*

*Also, we added a figure on the Ice thickness estimates for land-terminating glaciers and lake-terminating glaciers to the appendix (Fig. A5).*

**Minor comments**

L10: the term appears as "proglacial" in the title and "pro-glacial" here. Please be consistent with one use (I think the non-hyphenated version is preferable).

*Response:*

*We thank the referee and will use only "proglacial" in the revised manuscript.*

>*Solution: We changed this issue as suggested by the referee.*

L15: substantially more heterogeneity than what?

*Response:*

*Indeed 'than land-terminating glacier's. We will rewrite this sentence to create more clarity. We will write:*

*"We find that centre flow line velocities of lake-terminating glaciers are more than double those of land-terminating glaciers (18.8 vs 8.24 m yr⁻¹) and show substantially more heterogeneity than land-terminating glaciers around glacier termini."*

>*Solution:*

>*In line 15-16 we wrote: "We find that centre flow line velocities of lake-terminating glaciers are more than double those of land-terminating glaciers (18.8 vs 8.24 m yr-1) and show substantially more heterogeneity around glacier termini.".*

*We changed this to (line 15-17): "We find that centre flow line velocities of lake-terminating glaciers are more than double those of land-terminating glaciers (18.8(18.5 – 19.1) vs 8.24(8.17 – 8.35) m yr-1) and show substantially more heterogeneity than land-terminating glaciers around glacier termini."*

L16: effects > affects

*Response:*

*We will change this in the manuscript*

*Solution: We changed this in the manuscript.* L16: it is not clear what you are saying affects half of clean ice glaciers. Dynamic thinning? Terminal velocity heterogeneity?

*Response:*

*Indeed, this refers to dynamic thinning. We will clarify this in the revised manuscript. We will write:*

*"We attribute this large heterogeneity to the varying influence of lakes on glacier dynamics, resulting in differential rates of dynamic thinning, which causes about half of the clean-ice lake-terminating glacier population to accelerate at the glacier termini."*

> *Solution:*
>
> *We wrote in line 16-18: "We attribute this large heterogeneity to the varying influence of lakes on glacier dynamics, resulting in differential rates of dynamic thinning, which effects about half of the clean-ice lake-terminating glacier population."*
>
> *We changed this to 17-19: "We attribute this large heterogeneity to the varying influence of lakes on glacier dynamics, resulting in differential rates of dynamic thinning, which causes about half of the lake-terminating glacier population to accelerate at the glacier termini."*

L41: do you mean "to cause" dynamic thinning?

*Response:*

*We thank the referee his/her comment but think that 'through' might be more appropriate here.*

L54: what do you mean by "rapidly evolving environments"? Vague term that makes the meaning of this clause uncertain.

*Response:*

*We agree with the referee that we should be more specific here. We mean with this 'a state of a glacier that is far out of balance caused by environmental conditions (i.e., temperature) that are rapidly changing'. We will adapt the text for more clarity.*

> *Solution:*
>
> *We wrote (line 52-54): "Secondly, dynamical changes result from processes that act at the terminus and trigger a retreat and reduce along-flow resistive stresses (Nick et*

*al., 2009), which can be especially important in rapidly evolving environments (Benn et al., 2007b)."*

*We changed this to (line 55-58): "Secondly, dynamical changes result from processes that act at the terminus and trigger a retreat and reduce along-flow resistive stresses (Nick et al., 2009), which can be especially important in rapidly evolving environments (Benn et al., 2007b), such as the Himalayan region where the number and area of proglacial lakes are rapidly increasing."*

L55: what is partially decoupled from climate? The transition from land to lake terminating?

*Response:*

*Yes, we will rewrite this sentence to be clearer in the revised manuscript. We will write:*

*"In alpine settings, the transition from a land-terminating glacier to a lake-terminating glacier could therefore change the dynamic regime of the glacier, and such a transition might be partially decoupled from climate (Benn et al., 2012)."*

*Solution: We wrote (line 54-56): In alpine settings, the transition from a land-terminating glacier to a lake-terminating glacier could therefore change the dynamic regime of the glacier, something that might be partially decoupled from climate (Benn et al., 2012).*

*We changed this to (line 58-59): "In alpine settings, the transition from a land-terminating glacier to a lake-terminating glacier could therefore change the dynamic regime of the glacier, and such a transition might be partially decoupled from climate (Benn et al., 2012)."*

The Fig 1: I would suggest using the term "excluded" rather than "uncovered" because of "uncovered" sometimes being used synonymously with "clean" or "debris-free" in a debris covered glacier context.

*Response:*

*We thank the referee and follow his/her suggestion.*

*Solution: We changed this in the text and in the figure.*

L100: A reference like Anderson & Anderson, 2016 seems relevant here. Link: https://doi.org/10.5194/tc-10-1105-2016

*Response:*

*We thank the referee and follow his/her suggestion.*

*Solution: We added Anderson & Anderson, 2016 as a reference here.*

L138: "The maximum number of image pairs separated by one year was selected for the month of November, as this month is associated with low cloud cover and a relatively high snow line." – It is unclear what you mean by this. Where is a "maximum number" coming from this analysis? Are you saying you're using November as an end-of-year date?

*Response:*

*We use all available imagery from the month November. 'Maximum number' is indeed somewhat misplaced here and will be omitted in the revised manuscript.*

> *Solution: We thoroughly have rewritten section 3.1.2. Please see the revised manuscript for the (tracked) changes made.*

Sec 3.1.2 & Table 2 - we need more detail about what velocity fields represent? Rather than just "effective date", it would be useful to know the date of the first and second images used for each correlation. Or at least the midpoint date and the time span between the two images used. Otherwise we don't know if we are seeing annual velocities, seasonal velocities, or some combination. Perhaps this could be visualized as a plot showing the temporal distribution of image pairs for each footprint?

*Response:*

*We thank the referee for his/her useful suggestion. We will prepare a plot as suggested and add this to the appendix, as we feel that it might be less appropriate for the main text.*

> *Solution: We felt adding this plot would be interesting, but slightly outside the scope of this paper.*

L146-148: does the off-glacier used for estimating coregistraiton error have a similar aspect & slope distribution as the studied glaciers? If not (e.g., steep glaciers & flat area for uncertainty estimate), this error estimate may not be accurate.

*Response:*

*For the off-glacier area we used mountainous areas that likely do not show a lot of mass movement or creeping surfaces (see figure 2, see this supplement). Very steep slopes in high permafrost areas (around the glaciers) are not appropriate as we cannot expect this to be zero. Also, glacier ice surfaces, especially the ablation zone, do not show such these extreme slopes. Therefore, we think that the stable area chosen is adequately representative.*

> Solution: Please see the several changes made to the new, merged, section 3.1.3.

L290-291: I am a little confused by this because it seems like basal friction and effective pressure should depend on each other (not be independent as stated). For example, a bed with lower effective pressure will be more slippery (lower friction). Can you better justify this statement or better describe what this experiment is meant to test? If you're not changing sliding rates (through reduced basal traction) by altering lake level, then what exactly are you doing?

*Response:*

*Thank you for this comment. In our manuscript we wrote in line 49-50:*

*'Two key factors can be identified which make lake-terminating glaciers distinctively different from their land-terminating counterparts, namely the stresses at the bed and the terminus of the glacier.'*

*In our experiment we try to get insight in both of these factors. Firstly, we conduct an experiment where basal friction is effective pressure dependent, allowing for both stresses at the bed and the terminus of the glacier. Secondly, we perform the same experiment, but let*

*basal friction be independent of effective pressure, which then would depend on other factors such as drag from surface roughness. Although not necessarily realistic, this experiment enables us to separate the influence of the proglacial lake on the glacier velocity through either basal friction or forces at the terminus of the glacier.*

L292: exponent in As is not superscripted.

*Response:*

*We thank the referee for spotting this mistake.*

Sec 4.1: How does the absence/presence of debris cover affect velocity uncertainty? Are there systematic differences in debris cover between lake & land terminating glaciers?

*Response:*

*We have already partially answered the latter question and devoted some text on this in the discussion. We wrote in line 521-522:*

*"This, together with the over-representation of clean-ice glaciers in the lake-terminating glacier population (50 out of 70), explains a large part of the lake-land velocity contrast (Fig. 8a)."*

*We will analyse the uncertainty distribution among debris-covered glaciers and clean glaciers. We will incorporate this into the text if this analysis shows to be important.*

> *Solution: Although the suggestion was interesting and worth exploring, we found the outcome not worthwhile to cover in the manuscript.*

Sec 4.2: it seems like the most relevant thing here is whether there are systematic differences between mismatch land & lake terminating glaciers between your estimates & those in ITS_LIVE. If all of your velocities are faster than ITS_LIVE, that doesn't seem like that big of an issue because your study focuses on differences between these groups and is less concerned with absolute accuracy of speeds. However, there would be a problem if lake terminating glaciers are systematically fast biased and land terminating glaciers are slow biased. This analysis should be undertaken.

*Response:*

*We agree with the referee that is an important consideration. However, as Figure 4 shows, the largest differences in velocity estimates between the different datasets are observed away from the glacier termini, where any contrasts in methodology or imagery should be most apparent (relating to reference window). This suggests that there is no indication why, around this area of interest, lake-terminating glaciers would be positively biased and, which forms a part of the referee's argument, land-terminating glaciers would be negatively biased.*

Table 5: I think you mean ± 4.1 for lake terminating slope (written as 41).

*Response:*

*We thank the referee for spotting this mistake.*

L392: Do you mean "concurrently, IF a large fraction…"? Or are you saying that this is true?

*Response:*

*We are saying that this is true as this is observed from our results. However, we will go through this sentence again and try to create more clarity.*

L473: I think you mean 1 km, not 1 km2 ?

*Response:*

*We thank the referee and will correct this flaw.*

L517: This sentence is fairly awkward and it is hard to determine what you are trying to say.

*Response:*

*We will rewrite this section to create clarity.*

> *In line 517 – 520 we wrote:*
>
> *"Lake-terminating debris-covered glaciers can evolve from the median glacier size land-terminating glacier population, whereas lake-terminating clean-ice glaciers predominantly evolve from land-terminating glaciers that are relatively great in surface area. This, together with the over-representation of clean-ice glaciers in the lake-terminating glacier population (50 out of 70), explains a large part of the lake-land velocity contrast (Fig. 8a)."*
>
> *We changed this to (line 539-542):*
>
> *"As a result, lake-terminating debris-covered glaciers develop from the glaciers whose area is close to the median of the land-terminating glacier population, whereas lake-terminating clean-ice glaciers predominantly evolve from larger land-terminating glaciers. This, together with the over-representation of clean-ice glaciers in the total lake-terminating glacier population (50 out of 70), explains a large part of the lake-land velocity contrast (Fig. 8a)."*

*References:*

*Farinotti, D., Huss, M., Fürst, J. J., Landmann, J., Machguth, H., Maussion, F. and Pandit, A.: A consensus estimate for the ice thickness distribution of all glaciers on Earth, Nat. Geosci., 12(3), 168–173, https://doi.org/10.1038/s41561-019-0300-3, 2019.*

**List with changes**

- For the uncertainty estimates, we changed from providing standard errors (SE) to providing the IQR.
- We considered the importance of ice thickness by devoting a paragraph to this parameter in section 5.1 and by adding a figure of the ice thickness (Fig. A5) in the appendix.
- Improved the connection between the remote sensing and the modelling part by putting the modelling part into context of the broader objective of this study. This is done by rewriting the start of section 3.6.1 and 4.7.
- Merged sections 3.1.3, 3.1.4 and 3.1.5 into a single section named 'Image Processing' and removed the formula for the matching surface to make this section slightly less technical.
- To have a more robust flow for the uncertainty assessment, we reordered section 3.3 to section 3.5 by putting section 'Uncertainty of the Velocity Field' and section 'Glacier Group Uncertainty' right after each other.
- In section 3.1.3 (Image Processing) we more thoroughly considered the relevant choices made for the stable area selection as the referees requested.
- In section 3.1.2 (Image Pair Selection) more emphasis is put on explaining why only imagery from the month of November is selected.
- The importance of using a gaussian window to improve the quality of the velocity data (section 3.4) is evaluated in section 4.1.
- De variable application of the IQR is explicitly mentioned each time when relevant to avoid confusion about the interpretation of this measure.

- In 76 – 77 we wrote: "More specifically, we seek to investigate the attribution of lake-driven changes in the velocity field to dynamic thinning and investigate the role that debris cover plays on glacier-lake dynamics." We changed this to: "More specifically, we seek to investigate the attribution of changes in the velocity field to dynamic thinning and investigate the role that debris cover plays on glacier-lake dynamics."In line 106-107 we newly write: "In this study we only focus on glaciers with an area larger than 3 km2, compared to the 5 km2 previously utilised by Dehecq et al. (2019), which enables us to add substantially more glaciers to our dataset (Fig. 6b).". Here, we refer to Fig. 6b, where the added values of including smaller glaciers is clearly illustrated.
- On all relevant localities (table 2, line 148, 149, 236, 237 and 355) we changed 'effective date' for 'central date'.
- In line 151 we wrote: "Each pixel represents the orientation of intensity gradient at that pixel, making the method invariant to illumination change, which is a desired property for feature tracking algorithms." We changed this to: "Each pixel represents the orientation of intensity gradient in the x and y direction at that pixel, making the method invariant to illumination change, which is a desired property for feature tracking algorithms."

- In line 280 we wrote: "We then tuned the sliding parameter ($A_s$) such that the maximum up glacier velocity reaches a typical value of 50 m yr$^{-1}$ and found a value of $A_s = 2.5 \times 10^6$ Pa m$^{-2/3}$ s$^{1/3}$." We changed this to: "We then tuned the sliding parameter ($A_s$) such that the maximum velocity near the ELA of the larger clean-ice

lake-terminating glaciers reaches a typical value of 50 m yr$^{-1}$ (Dehecq et al., 2019a; Gardner et al., 2020; Pronk et al., 2021), and found a value of A$_s$ = 2.5 × 10$^6$ Pa m$^{-2/3}$ s$^{1/3}$."

- We changed (line 517 – 520): Lake-terminating debris-covered glaciers can evolve from the median glacier size land-terminating glacier population, whereas lake-terminating clean-ice glaciers predominantly evolve from land-terminating glaciers that are relatively great in surface area. This, together with the over-representation of clean-ice glaciers in the lake-terminating glacier population (50 out of 70), explains a large part of the lake-land velocity contrast (Fig. 8a). Into (line 539-542):
"As a result, lake-terminating debris-covered glaciers develop from the glaciers whose area is close to the median of the land-terminating glacier population, whereas lake-terminating clean-ice glaciers predominantly evolve from larger land-terminating glaciers. This, together with the over-representation of clean-ice glaciers in the total lake-terminating glacier population (50 out of 70), explains a large part of the lake-land velocity contrast (Fig. 8a)."

- In line 15-16 we wrote: "We find that centre flow line velocities of lake-terminating glaciers are more than double those of land-terminating glaciers (18.8 vs 8.24 m yr-1) and show substantially more heterogeneity around glacier termini.". We changed this to (line 15-17): "We find that centre flow line velocities of lake-terminating glaciers are more than double those of land-terminating glaciers (18.8(18.5 – 19.1) vs 8.24(8.17 – 8.35) m yr-1) and show substantially more heterogeneity than land-terminating glaciers around glacier termini."

- We wrote in line 16-18: "We attribute this large heterogeneity to the varying influence of lakes on glacier dynamics, resulting in differential rates of dynamic thinning, which effects about half of the clean-ice lake-terminating glacier population."
We changed this to 17-19: "We attribute this large heterogeneity to the varying influence of lakes on glacier dynamics, resulting in differential rates of dynamic thinning, which causes about half of the lake-terminating glacier population to accelerate at the glacier termini."

- We wrote (line 52-54): "Secondly, dynamical changes result from processes that act at the terminus and trigger a retreat and reduce along-flow resistive stresses (Nick et al., 2009), which can be especially important in rapidly evolving environments (Benn et al., 2007b)."
We changed this to (line 55-58): "Secondly, dynamical changes result from processes that act at the terminus and trigger a retreat and reduce along-flow resistive stresses (Nick et al., 2009), which can be especially important in rapidly evolving environments (Benn et al., 2007b), such as the Himalayan region where the number and area of proglacial lakes are rapidly increasing."

- We wrote (line 54-56): In alpine settings, the transition from a land-terminating glacier to a lake-terminating glacier could therefore change the dynamic regime of the glacier, something that might be partially decoupled from climate (Benn et al., 2012).
We changed this to (line 58-59): "In alpine settings, the transition from a land-terminating glacier to a lake-terminating glacier could therefore change the dynamic regime of the glacier, and such a transition might be partially decoupled from climate (Benn et al., 2012)."

- In line 517 – 520 we wrote:

"Lake-terminating debris-covered glaciers can evolve from the median glacier size land-terminating glacier population, whereas lake-terminating clean-ice glaciers predominantly evolve from land-terminating glaciers that are relatively great in surface area. This, together with the over-representation of clean-ice glaciers in the lake-terminating glacier population (50 out of 70), explains a large part of the lake-land velocity contrast (Fig. 8a)."

We changed this to (line 539-542):

"As a result, lake-terminating debris-covered glaciers develop from the glaciers whose area is close to the median of the land-terminating glacier population, whereas lake-terminating clean-ice glaciers predominantly evolve from larger land-terminating glaciers. This, together with the over-representation of clean-ice glaciers in the total lake-terminating glacier population (50 out of 70), explains a large part of the lake-land velocity contrast (Fig. 8a)."

- For changes that need more context from the referee their comments, please see the indented proposed solutions beneath many of their comments.

---

## Referee Report (RR1)

**Re-review of "Proglacial Lakes Elevate Glacier Surface Velocities in the Himalayan Region" (tc-2021-90) by Pronke et al.**

**Overall:** The authors do a thorough job incorporating suggestions from myself at the other reviewer. I think this manuscript is near ready publication, but have a few comments remaining on the text on Lines 594 – 609 & Figure A5 relating to ice thickness differences between the land- and lake-terminating glaciers that should first be addressed. When published, I think this work will be of great interest to the community & I look forward to citing it!

I agree that there is some uncertainty pertaining to the Farinotti et al. (2019) ice thickness dataset, but it is the best resource we have for estimating this important variable.

A few ideas for clarifying & better supporting your claim about the importance of ice thickness:

1) Can you provide more quantitative data on the ice thickness difference, rather than just using qualitative terms (e.g., "substantial factor") and pointing to Fig. A5? For example, by what factor (e.g., 1.5x) thicker are lake-terminating glaciers than land-terminating glaciers? If you used a simple estimate like the shallow ice approximation, what velocity difference would you expect? What fraction of the observed velocity difference could this explain?

2) The fact that Fig 6b shows substantially faster flow for lake-terminating glaciers within the same area class suggests that variables in addition to ice thickness are driving the speed differences (because thickness generally scales with area). This may be worth pointing to to better support your claim of ice thickness differences not driving your observation.

3) It is not clear to me what you mean by "Concurrently, it is worth considering that the difference in ice thickness between land-terminating and lake-terminating glaciers is also due to the very presence of a lake". This seems like a "chicken-or-the-egg" problem – is ice thickness different there because the lake exists, or does the lake exist because ice thickness (and/or subglacial/proglacial topography) differ? Lakes are found in basins (to state the obvious), and basins/overdeepenings also promote thick/fast ice. We wrestled a lot with covariance between ice thickness and other variables in the context of proglacial lakes in the paper referenced below (particularly in Secs 4.4 & 5.4). It may be worth reading and referencing those ideas.

Field, H. R., Armstrong, W. H., and Huss, M.: Gulf of Alaska ice-marginal lake area change over the Landsat record and potential physical controls, The Cryosphere, 15, 3255–3278, https://doi.org/10.5194/tc-15-3255-2021, 2021.

4) In Lines 605-607, I don't think you can say the link is causal because of the above-referenced "chicken-or-egg" issue. You show an association between lakes & faster velocity, but not necessarily lakes causing faster velocity.

Review by William Armstrong

---

## Author Response (AR2)

**Response by Jan Pronk and others to William Armstrong his comment on tc-2021-90.**

We would like to thank the referee for his second look on our manuscript. His comments, that had a specific focus on a more thorough consideration of ice thickness estimates, helped to improve our work considerably. As a result, current major changes to the manuscript are largely limited to the paragraph related to the ice thickness differences (lines 590 - 612 & Figure A5). Below we provide our response to each comment.

1) Can you provide more quantitative data on the ice thickness difference, rather than just using qualitative terms (e.g., "substantial factor") and pointing to Fig. A5? For example, by what factor (e.g., 1.5x) thicker are lake-terminating glaciers than land-terminating glaciers? If you used a simple estimate like the shallow ice approximation, what velocity difference would you expect? What fraction of the observed velocity difference could this explain?

**Response:**

We agree with the referee that providing more quantitative data on the ice thickness difference would improve this relevant section, and we therefore decided to adopt his suggestions. For this, we followed the approach of Dehecq et al. (2019), where a first order approximation is made to relate changes in surface velocity to changes in driving stress. Assuming all factors other than ice thickness changes remain constant, we relate all changes in driving stress to changes in ice thickness:

$$1 + \frac{\partial U_s}{U_s} = (1 + \frac{\partial H}{H})^4$$

This gives a rough estimate of how many of the observed velocity change can be explained by changes in ice thickness, when no other factors (such as a lake or slope) play a role. We solely apply this approximation to the whole ablation zone, as this approximation is only appropriate in the limit of  $\frac{\partial H}{H}$ <1. Finding a mean median ice thickness of lake-terminating glaciers of ~110m and land-terminating glaciers of ~100m over the ablation zone, we find that only 28% to 64% of the observed velocity contrast can be explained by this approximation.

2) The fact that Fig 6b shows substantially faster flow for lake-terminating glaciers within the same area class suggests that variables in addition to ice thickness are driving the speed differences (because thickness generally scales with area). This may be worth pointing to to better support your claim of ice thickness differences not driving your observation.

Response:

**We thank the referee for this suggestion and adopted this in the manuscript.**

3) It is not clear to me what you mean by "Concurrently, it is worth considering that the difference in ice thickness between land-terminating and lake-terminating glaciers is also due to the very presence of a lake". This seems like a "chicken-or-the-egg" problem – is ice thickness different there because the lake exists, or does the lake exist because ice thickness (and/or subglacial/proglacial topography) differ? Lakes are found in basins (to state the obvious), and basins/overdeepenings also promote thick/fast ice. We wrestled a lot with covariance between ice thickness and other variables in the context of proglacial lakes in the paper referenced below (particularly in Secs 4.4 & 5.4). It may be worth reading and referencing those ideas.

**Response:**

We agree with the referee that this raised a "chicken-or-the-egg" issue, and therefore decided to remove this argumentation from our manuscript. Also, we gladly added the interesting study from Field et al. (2021) to our discussion (lines 627 - 628).

4) In Lines 605-607, I don't think you can say the link is causal because of the above-referenced "chicken-or-egg" issue. You show an association between lakes & faster velocity, but not necessarily lakes causing faster velocity.

**Response:**

We agree with the referee and would like to thank him for pointing this out. In line with the previous comment form the referee, with decided to not argue that this link is causal.

---

## Author Response (AR3)

**Response by Jan Pronk and others to Daniel Farinotti his comment on tc-2021-90.**

We would like to thank the editor for his look on our almost finished manuscript. His comments we mainly focusing on improving notating, logic, and flow, which helped to improve the text considerably. Also, we decided to change, in line with the editor's and referee's suggestion, the title to: 'Contrasting Surface Velocities Between Lake- and Land-Terminating Glaciers in the Himalayan Region'. Please see the specific comments for all the changes made.

- Line 16-17: Rewrote the sentence to give more clarity on the numbers. Also, made from here onwards the number of significant digits consistent throughout the text.
- Line 19: Changed to 'accelerate towards'
- Line 20: We followed the editor's suggestion, please see the manuscript for the changes.
- Line 121: We changed 'coverage' to 'percentage of glacierised area covered'.
- Line 160: It is possible to get a non-November central date with just November imagery. For example, lets assume you've got one image from Nov. 2016 and one form Nov. 2017, you'll find a central date of June 2017. However, to gain clarity, we decided to remove the 'central date' column and elaborate on the date range in the caption.
- Line 171: We thank the editor for this comment and changed 'Iteration Step' to 'Iteration Stepsize'. This is simply the step the algorithm 'jumps' to the next spot to calculate the new velocity again.
- Line 180: We removed the part 'in addition to the glacierised areas' to improve the flow of this line.
- Line 200: We added to line 201: 'The number *1.483* is a scale factor that relates the MAD to the standard deviation.'
- Line 278: We thank the editor for spotting this mistake, and corrected this accordingly in the new manuscript.
- Line 303: Added 'which is on the low side compared to values used at marine terminating outlet glaciers (i.e., Enderlin et al., 2013; Nick et al., 2009).'
- Line 372: Removed '$g_c$' from the manuscript.
- Line 384: Changed this to: "(median: 18.83 m yr$^{-1}$; IQR uncertainty range: 17.55 to 18.06 m yr$^{-1}$)". And kept the older, shorter, notation the same thereafter.
- Line 395: Removed all the '-' signs in front of the slope values in the manuscript.
- Line 406: We thank the editor for his suggestion and adopted those in the manuscript.
- Line 412: Changed this to 'at those northwards flowing glaciers'
- Line 448: We agree with the editor that some clarification was needed, and rewrote the relevant sentence accordingly (please see the manuscript).
- Line 454: We removed the reference to Matplotlib and Python from the manuscript.
- Line 534 – 540: We agree that overall some improvement was needed here. We therefore followed the suggestions from the editor and restructured sentences when needed (please see the manuscript).
- Line 549: We changed this into 'heterogeneity in velocity'.
- Line 555: We changed this into 'acceleration towards the termini'.
- Line 558: We agree with the editor and changed 'the frontal boundary condition' into 'changes in the force balance at the glacier termini'.
- Line 581: We thank the editor for spotting this mistake and changed this into 'is indicative for'.
- Line 613 – 626: We agree that this section especially needed some structuring as it appeared to be a little 'jumpy' with contradicting statements. Please see this relevant section with the improved flow on the importance of ice-thickness.
- Line 635: We thank the editor for spotting this mistake and removed 'Himalayan' in the new manuscript.

- Line 639: We clarified this line by changing 'for all these rapid changes' into 'With such a feedback mechanism'.
- Line 669: We followed the editor's suggestion and removed the spectral details from the manuscript.

Best regards,

Jan Pronk & others.